# CONCEPT-GUIDED DICTIONARY LEARNING FOR INTERPRETABLE CONCEPT EXTRACTION AND ATTRIBUTION IN LARGE VISION–LANGUAGE MODELS

## ABSTRACT

Autoregressive large vision–language models (LVLMs) generate text sequentially, conditioning each token on evolving multimodal states. This makes it difficult to assess whether predictions are grounded in **visual concepts** or instead reflect hallucination or bias. Existing concept-discovery approaches, such as TCAV, CRAFT, and CLIP-Dissect, are designed for encoder-only or contrastive models. At the same time, recent LVLM methods such as CoX-LMM depend on labeled concepts and simplified settings, limiting scalability.

We propose **Concept-Guided Dictionary Learning (CGDL)**, a weakly supervised and scalable framework for discovering multimodal concept vectors in autoregressive LVLMs. CGDL first prompts the model to identify textual concepts in a dataset. For each concept, it constructs positive and negative patch sets from SAM-generated foreground crops and randomized background patches. Next, we apply a binary prompt to extract hidden activations for positive and negative patches. A contrastive dictionary learning stage then disentangles concept-aligned activations from residual noise, yielding sparse, monosemantic vectors that reveal **semantically aligned visual–textual interactions** and enable faithful attribution of predictions to visual evidence.

On **ImageNet-1k** and **MSCOCO**, CGDL outperforms recent interpretability methods with up to **4% higher sparsity**, **11% greater stability**, **17% lower overlap**, and strong attribution faithfulness, while scaling efficiently to large concept vocabularies. These results advance concept-based interpretability for LVLMs and provide a practical step toward transparent multimodal reasoning.

## 1 INTRODUCTION

Large vision–language models (LVLMs) generate text *autoregressively*, conditioning each token on evolving multimodal states. This enables rich, context-sensitive prediction but raises unique interpretability challenges: predictions may stem from spurious correlations or hallucinations rather than *visual evidence*. Existing methods, such as saliency maps or training-based grounding, localize objects but do not reveal the higher-level *concepts* driving model decisions. Concept-based explanations offer more conceptual insights, yet most methods assume static feature spaces. Approaches like *TCAV* Kim et al. (2018), *CRAFT* Fel et al. (2023b), *CLIP-Dissect* Oikarinen & Weng (2023), and *Holistic* Fel et al. (2023a) cannot extend to autoregressive models and are restricted to single-object settings, while *CoX-LMM* Parekh et al. (2024) additionally requires a labeled concept token, further limiting scalability (Table 1).

We propose **Concept-Guided Dictionary Learning (CGDL)**, a weakly supervised framework for scalable concept discovery in autoregressive LVLMs. We use native vision–language models (e.g., Qwen2.5-VL and Gemma-3n Team (2025b;a)), rather than models that use a frozen LLM adapted via a bridging mechanism Vallaeys et al. (2024), because this setting lets us trace how image concepts propagate through an autoregressively trained VLM all the way to the final layer. CGDL finds candidate concepts and treats each one as a one-vs-all representation learning problem with a two-basis decomposition. It constructs positive and negative patch sets using segmentation and random cropping, prompts the model with a binary question to identify whether the concept exists in each crop, and extracts the model's hidden activations for both positive and negative patches. This formulation forces the dictionary to disentangle activations from residual noise (negative patches), yielding sparse, non-overlapping vectors that faithfully align visual and textual modalities.

Table 1: Comparison of concept-based interpretability methods. Only *CGDL* supports weakly supervised, scalable discovery in autoregressive LVLMs; most prior methods target encoder-only image-encoder (IE) models and single-object concepts, and they cannot perform text grounding (TG).

| Method | Type | TG | AutoReg | Limitation/ Advantage |
|---|---|---|---|---|
| TCAV Kim et al. (2018) | IE | ✗ | ✗ | Supervised |
| ACE Ghorbani et al. (2019) | IE | ✗ | ✗ | Single object |
| CRAFT Fel et al. (2023b) | IE | ✗ | ✗ | Single object |
| EAC Sun et al. (2023) | IE | ✗ | ✗ | Single object |
| CLIP-Dissect Oikarinen & Weng (2023) | IE | ✓ | ✗ | Single object |
| Holistic Fel et al. (2023a) | IE | ✗ | ✗ | Single object |
| CoX-LMM Parekh et al. (2024) | AutoReg | ✓ | ✓ | Single object |
| MCD Grobrügge et al. (2025) | IE | ✗ | ✗ | Single object |
| **CGDL (Ours)** | AutoReg | ✓ | ✓ | / Unlimited-object |

While most concept extraction methods rely on similar dictionary learning formulations, prior work has shown that the quality of discovered concepts depends more on how the data is presented to the dictionary than on the specific factorization algorithm Grobrügge et al. (2025); Sun et al. (2023). CGDL is not merely *Semi-NMF* Fel et al. (2023a) with preprocessing; rather, it provides a general, model-agnostic framework that reframes concept discovery as a contrastive one-vs-all decomposition. The two-basis design prevents atom collapse in large vocabularies, allowing CGDL to scale robustly to thousands of concepts Kim & Park (2008). By combining weakly supervised concept bags, weak localization, residual contrast, and two-basis factorization, CGDL achieves monosemantic, multimodal vectors at scale—something previous dictionary learning methods for LVLMs have not demonstrated.

**Our contributions are:**

- We introduce *CGDL*, a weakly supervised framework for scalable concept discovery in autoregressive LVLMs, overcoming the single-object limitation of prior methods.

- We leverage the *Segment Anything Model (SAM)* Kirillov et al. (2023) for weak concept localization, providing high-quality positive/negative patch sets.

- We propose a contrastive residual extraction scheme that enforces monosemanticity by separating concept vs. no-concept activations.

- We show that CGDL yields faithful multimodal attribution, bridging visual and textual modalities using extensive experiments on **ImageNet-1k** and **MSCOCO**, demonstrating superior sparsity, stability, and faithfulness over prior methods, scaling to 1k concepts with improved attribution quality (up to **+9% CLIPScore** (Radford et al., 2021a), **+4% BERTScore** (Devlin et al., 2019)).

Together, these advances provide a practical step toward transparent multimodal reasoning, bridging the gap between autoregressive generation and human-understandable concept-based explanations.

## 2 RELATED WORK

A central goal in interpretability is to uncover how internal representations encode semantically meaningful concepts. Early work introduced Concept Activation Vectors (CAVs) (Kim et al., 2018), which quantify model sensitivity to human-defined concepts but require curated examples, limiting scalability. Subsequent extensions sought to automate discovery (Ghorbani et al., 2019), yet dependence on segmentation quality hindered robustness.

Alternative approaches use saliency maps (Selvaraju et al., 2017; Strumbelj & Kononenko, 2014) or training-based grounding (Kang et al., 2024; Zhang et al., 2023; Ma et al., 2024) to highlight *where* evidence lies. While effective for localization, these methods do not articulate *what* abstract concepts the model internally represents, thereby complementing, but not replacing, concept-based explanations. To reduce supervision, some methods decompose hidden states into interpretable factors. NMF (Liu et al., 2025), dictionary learning with prototypes (CRAFT) (Fel et al., 2023b), PCA, UMAP, and sparse autoencoders (Pach et al., 2025) have shown promise in vision-only settings,

with unifying benchmarks (Fel et al., 2023a). CLIP-based concept attribution (Oikarinen & Weng, 2023; Dreyer et al., 2025) further advanced automated discovery, but these methods remain restricted to vision encoders.

Moreover, concept bottleneck models (CBMs) explain vision classifiers or image encoders by using an LVLM to extract textual concepts from images and then training a linear proxy on these concepts to predict the model output (Oikarinen et al., 2023; Yang et al., 2023). We also use an LVLM to first identify which concepts are present in the dataset, but we do not train any proxy model. Instead, the discovered concepts guide the extraction of concept vectors from the same LVLM's hidden states for post hoc explanation of that LVLM.

Feature finding (Pach et al., 2025; Zhang et al., 2025) for understanding a model's knowledge and for model steering using sparse autoencoders is another promising direction for concept-level explanation. However, these methods integrate a sparse autoencoder (SAE) inside the model, which changes the architecture and can alter its behavior, making them less suitable for post hoc local explanation. While concept extraction overlaps with mechanistic interpretability (Templeton, 2024; Pach et al., 2025; Elhage et al., 2022), our focus is on *single-layer monosemantic concept discovery* in autoregressive LVLMs rather than neuron- or circuit-level analysis. Alternative approaches such as attention maps (Jain & Wallace, 2019), causal tracing (Meng et al., 2023), and linear probes (Alain & Bengio, 2018) provide useful insights but suffer from weak causal grounding, poor scalability, or reliance on labeled supervision.

Autoregressive LVLMs present new challenges: activations evolve across time steps, residual streams encode multiple dependencies, and concepts rarely align with a single hidden state (Templeton, 2024). CoX-LMM (Parekh et al., 2024) adapted Semi-NMF (SNMF) (Trigeorgis et al., 2014) to LVLM activations but struggles with (i) reliance on tokenized object names, (ii) limited support for multi-token concepts, (iii) background noise from full-image extraction, and (iv) persistent polysemanticity in residual streams. While methods such as TCAV, CRAFT, and CLIP-Dissect pioneered concept-based interpretability, they target static encoders and cannot be applied to LVLMs. CoX-LMM incorporates many of these ideas into a dictionary learning framework for autoregressive models and thus serves as the most representative baseline. We therefore compare CGDL against CoX-LMM using its strongest dictionary learning variants (SNMF, SAE) to ensure a fair comparison.

Unlike CoX-LMM and other concept extraction methods that rely on labeled tokens and often yield polysemantic vectors and assume single-object settings, we propose *Concept-Guided Dictionary Learning (CGDL)*. CGDL introduces contrastive residual extraction with spatially localized image crops and candidate textual concepts, enforcing a clean separation between *concept* and *noise*. This produces faithful, monosemantic concept vectors. To the best of our knowledge, CGDL is the first framework to scale weakly supervised concept discovery from single-object to multi-object settings.

## 3 PRELIMINARIES

Recent work on concept vector extraction (Ghorbani et al., 2019; Sun et al., 2023; Fel et al., 2023a; Parekh et al., 2024) shows that many of these methods can be framed as *dictionary learning*, where activations are approximated by a small set of interpretable vectors. Formally, given activations $S \in \mathbb{R}^{n \times d}$ with $n$ samples and $d$-dimensional features, we posit $K$ latent concepts and solve

$$\underset{U \in \mathbb{R}^{d \times K}, \, V \in \mathbb{R}^{n \times K}}{\arg \min} \|S - VU^\top\|_F^2,$$

where $U = [u_1, \ldots, u_K]$ are the **concept vectors** (CAVs) and $V = [v_1^\top ; \ldots ; v_n^\top]$ are the **activations**, with row $v_i$ giving concept coordinates of sample $i$.

This factorization unifies prior approaches via constraints on $(U, V)$:

$$
\begin{cases}
v_i \in \{e_1, \ldots, e_K\}, & \text{K-Means (ACE) Ghorbani et al. (2019),} \\
U^\top U = I, & \text{PCA Graziani et al. (2023),} \\
S \geq 0, \, U \geq 0, \, V \geq 0, & \text{NMF (CRAFT Fel et al. (2023a;b), ,} \\
U \text{ free}, \, V \geq 0, & \text{Semi-NMF (SNMF) Trigeorgis et al. (2014); Parekh et al. (2024),} \\
V = \psi(S), \, \|v_i\|_0 \leq s, & \text{Sparse Autoencoder (SAE) Templeton (2024); Pach et al. (2025).}
\end{cases}
$$

Columns of $U$ are concept vectors (CAVs), rows of $V$ are per-sample activations. Special cases include PCA (orthogonal bases), NMF (nonnegative factors), SNMF (mixed-sign bases with nonnegative activations), K-Means (one-hot codes), and SAE (encoder-decoder with sparsity).

## 4 METHOD

We begin by formalizing LVLMs as black-box systems with accessible intermediate activations, taking multimodal inputs (text, image) and producing corresponding outputs (text/ set of tokens).

### 4.1 POSITIVE AND NEGATIVE CONCEPTS

We address the limitations of single-object dependence and polysemantic behavior in CoX-LMM by introducing *concept-example bags*—collections of image patches that serve as positive (concept-present) or negative (concept-absent) instances for each automatically discovered concept $c_k \in \{c_1, \ldots, c_K\}$.

Given an unlabeled image set $\mathcal{I} = \{I_k\}_{k=1}^N$, the LVLM $f$ predicts candidate concepts for each image using a structured prompt (See Appendix B for details): $C(I_k) = f(I_k, \texttt{prompt}) \subseteq \mathcal{V}$ and the global vocabulary is

$$\mathcal{C} = \bigcup_{k=1}^N C(I_k).$$

For each $c \in \mathcal{C}$, we collect supporting images $\Phi(c) = \{I_k \mid c \in C(I_k)\}$. Using SAM Kirillov et al. (2023), a patch operator $\mathcal{P}(I_k, c) \in \{\mathrm{randomcrop}(I_k), \mathrm{sam}(I_k, c)\}$ localizes the region associated with $c$. The recent visually grounded CBM method Prasse et al. (2025) uses SAM Kirillov et al. (2023) for segmentation-based concept crops to train and explain CBM outputs. In contrast, we leverage SAM to construct concepts for LVLMs and explain LVLM behavior post hoc. Because such patches may include background context, the resulting *concept-example bag* is a mixture of positives and negatives:

$$\mathcal{B}(c) = \{ \mathcal{P}(I_k, c) : I_k \in \Phi(c) \} = \mathcal{B}^+(c) \cup \mathcal{B}^-(c).$$

Unlike prior approaches that rely on annotated single-object images, this formulation is *weakly supervised* and scales to multi-object datasets. For example, the concept "stripes" may emerge from zebras, tigers, or cats, without requiring manual concept labels.

### 4.2 CONCEPT-GUIDED DICTIONARY LEARNING

According to Fel et al. (2023a), most prior concept-expansion methods rely on dictionary learning for concept extraction; CoX-LMM is no exception. The key difference lies in the data passed to the dictionary learning algorithm, which critically affects concept quality Grobrügge et al. (2025); Sun et al. (2023).

Unlike prior approaches for concept extraction in LVLMs that rely on open-ended token generation and single-token analysis, our method restricts token generation to reduce noise and entanglement. Open-ended generation makes a single token's hidden representation noisy, since each token is influenced by the entire sequence. This leads to overlapping concepts Templeton (2024) and prevents dictionary elements from representing clean, disentangled semantics. Multi-object images further exacerbate this effect, as activations mix features from different objects.

We address these issues with *Concept-Guidance*, a contrastive residual extraction scheme. The model is prompted to output either the target concept $c_k$ or $\mathrm{No}{-}c_k$, ensuring that activations are aligned with a single concept and enforcing monosemanticity. This binary design yields cleaner, concept-focused residuals.

```
prompt_cg = ``Does the image contain c_k?  If yes, output
c_k; otherwise, output No-c_k.''
```

For each concept bag $\mathcal{B}(c_k)$, we query the model with the above prompt and collect residual activations. Following Koh et al. (2020); Alam et al. (2025), we decompose the LVLM into an **embedding function** $g$ (vision encoder, bridging, decoder attention) and an **output function** $h$ (projection and softmax). Based on Fel et al. (2023a); Parekh et al. (2024), we analyze the penultimate residual

layer (language_model.norm). For an image crop $x^{c_k} \in \mathcal{B}(c_k)$ and cached prefix $\hat{y}_{<t}$, the embedding function produces

$$a_t^{(l)} = g^{(l)}(x^{c_k}, \texttt{prompt}_{cg}, \hat{y}_{<t}) \in \mathbb{R}^p, \tag{1}$$

where $p$ is the residual dimension Geva et al. (2020). The output function then predicts

$$\hat{y}_t = h^{(l)}(a_t^{(l)}). \tag{2}$$

According to Geva et al. (2020), we can average residuals across tokens in each response to obtain a concept-level embedding without losing semantic meaning.

$$s_m = \tfrac{1}{T_m} \sum_{t=1}^{T_m} a_t^{(l)} \in \mathbb{R}^p, \tag{3}$$

where $T_m$ is the number of generated tokens for sample $m$. Collecting $M$ such samples yields

$$S = [s_1, \ldots, s_M] \in \mathbb{R}^{M \times p}, \tag{4}$$

which serves as input for concept extraction. Unlike approaches that rely solely on token embeddings, this formulation captures information from multi-token concepts (e.g., *hot dog*) rather than splitting them into isolated tokens (*hot*, *dog*).

Dictionary learning then decomposes $S$ into concept and negation bases:

$$S \approx VU^T, \quad V \in \mathbb{R}^{M \times 2}, \; U \in \mathbb{R}^{p \times 2}.$$

Here, $U$ contains basis vectors for $c_k$ and No-$c_k$, while $V$ captures sample activations. When restricting each bag to two bases (concept vs. negation), the per-bag cost reduces to $\mathcal{O}(M_c p + M_c^2)$, where $M_c$ is the number of samples in $\mathcal{B}(c_k)$. Over $K$ concepts, the total complexity scales as $\mathcal{O}(K(M_c p + M_c^2))$, which remains efficient for large concept sets, consistent with the per-iteration complexity of Semi-NMF Kim & Park (2008); Ding et al. (2010). This contrasts with full NMF, where larger dictionary sizes ($K \gg 2$) lead to cubic dependence on $K$, and with Sparse Autoencoders (SAE), where training requires backpropagation through millions of parameters. By reducing each bag to two bases, our approach avoids interference among dictionary atoms and remains scalable for LVLM interpretability.

This formulation contrasts each concept against all others (akin to one-vs-all classification), capturing inter-concept relations without requiring oversized dictionaries. Unlike prior methods that entangle concepts across long sequences, Concept-Guidance yields disentangled, monosemantic residuals suitable for large-scale LVLM interpretability.

### 4.3 POSITIVE CONCEPT VECTOR IDENTIFICATION

Inspired by the visual grounding in Parekh et al. (2024), for each feature $u \in U$ from the dictionary decomposition, we extract the *maximum activated crops (MAC)*—the top-$\alpha_{\text{MAC}}$ image patches that most strongly activate it, i.e., the highest values in the $k$-th column of the activation matrix $V$:

$$X_{k,\text{MAC}} = \{i \mid v_i^{(k)} \text{ is among the top-}\alpha_{\text{MAC}} \text{ values of } v^{(k)}\}.$$

Since each concept bag contains both $c_k$ (positive) and $\texttt{No}-c_k$ (negative) samples, we select the basis as

$$k^* = \arg \max_{j \in \{0,1\}} \left| \{i \in X_{k,\text{MAC}} \mid \hat{y}_i = c_k\} \right|,$$

i.e., the one aligned with the majority of positive outputs. We then define $u_{k^*} \in \mathbb{R}^p$ as the concept vector and $X_{k^*,\text{MAC}}$ as its supporting crops. Repeating this across all concept bags yields a dictionary $U = [u_1, \ldots, u_K] \in \mathbb{R}^{p \times K}$ with visual groundings $X_{\text{MAC}} = \{X_{1,\text{MAC}}, \ldots, X_{K,\text{MAC}}\}$.

For textual grounding, we use the LVLM's output function directly. Recall that for a residual activation $a_t^{(l)}$, the model predicts the next token as

$$y_t = h^{(l)}(a_t^{(l)}). \tag{2}$$

Analogously, for each concept feature $u_k$, we compute its token distribution by applying the same output function:

$$q_k = h^{(l)}(u_k) \in \mathbb{R}^{|V|}$$

where $|V|$ denotes the size of the model's output vocabulary (i.e., the number of distinct tokens in the LVLM). We then select the top-$\tau$ tokens (e.g., 50) with the highest scores, remove stopwords and noise, and define the resulting set as $X_{k,\text{MAT}}$ for concept $u_k$. Collecting across all concepts yields $X_{\text{MAT}} = \{X_{1,\text{MAT}}, \ldots, X_{K,\text{MAT}}\}$.

### 4.4 Concept Attribution and Multi-modal Alignment

By taking motivation from Kim et al. (2018), we project residual activations at layer $l$ onto the learned concept subspace $\widehat{U} = [u_1, \ldots, u_K] \in \mathbb{R}^{p \times K}$. At the token level, concept scores are

$$\alpha_j^{(t)} = \cos\_sim(u_j, a_t^{(l)}),$$

while at the phrase level, we use the mean-pooled activation $s_m = \frac{1}{T_m} \sum_{t=1}^{T_m} a_t^{(l)}$ to compute

$$\alpha_{m,j} = \cos\_sim(u_j, s_m).$$

Thus $a_t^{(l)}$ provides fine-grained token attribution, whereas $s_m$ captures phrase/sentence-level semantics. The most activated concept is

$$k^* = \arg\max_j \alpha_{m,j},$$

and its groundings $X_{\text{MAC}}[k^*]$ (visual) and $X_{\text{MAT}}[k^*]$ (textual) provide multimodal explanations.

In the *text-only mode*, this procedure further allows us to assess whether purely textual prompts activate the same feature directions as multi-modal inputs, thereby quantifying *multi-modal alignment*. This shared projection space links visual and textual semantics, supporting faithful cross-modal interpretability.

## 5 Experiments

### 5.1 Model and Data

We evaluate three recent instruction-tuned LVLMs—**Qwen2-VL-7B** Wang et al. (2024), **Qwen2.5-VL-7B** Team (2025b), and **Gemma-3n-E4B** Team (2025a)—keeping all models *frozen* to ensure post hoc interpretability. Results for the two Qwen models are reported in Appendix E.

For concept learning, we collect **300 examples per class** from ImageNet and MSCOCO, extracting features from the *penultimate norm* layer, shown to yield high-quality embeddings (Parekh et al., 2024; Kim et al., 2018). Evaluation uses a **disjoint set of 50 images per class** from the validation splits of ImageNet (1,000 classes) and MSCOCO (10 randomly selected objects).

Unlike CoX-LMM, which requires **ImageNet** labels and **MSCOCO** caption tokens during extraction, CGDL is weakly supervised: we collect a large pool of concept examples and retain the top 1,000 most frequent concepts for ImageNet and the top 10 for MSCOCO, with each concept bag capped at 1,600 cropped patches. This setup allows us to study scalability across very different concept set sizes, while keeping comparisons fair against CoX-LMM, which requires ground-truth labels for multiple concepts. Images are resized to 500 pixels in width and cropped into $200 \times 200$ windows with 0.2 overlap, yielding on average 5–6 crops per image. This makes the effective number of samples per bag comparable to the 300 full images used in CoX-LMM.

We compare **CGDL** against CoX-LMM Parekh et al. (2024) using two dictionary learning methods: Sparse Autoencoders (SAE) Pach et al. (2025) and Semi-Nonnegative Matrix Factorization (SNMF) Trigeorgis et al. (2014). We also include a simple baseline, **SIMPLE**, which tests whether residuals with the highest $l_2$-norm align with concepts.

Ground-truth concepts are defined by image class labels, with correctness measured by top-1 alignment. For faithfulness testing, we evaluate on the same 10 MSCOCO objects.

### 5.2 Evaluation Metrics

Concept discovery can be framed as a special case of dictionary learning. Following the evaluation protocol of Fel et al. (2023a), we first assess the quality of discovered features using three standard metrics: **Sparsity** ($\uparrow$), **Stability** ($\downarrow$), and **Overlap** ($\downarrow$). Based on these results, SNMF emerges as the most suitable method for downstream evaluation.

Scalability is evaluated with both 10 and 1,000 concepts, demonstrating that purity and uniqueness are maintained as the number of concepts grows. Attribution quality Parekh et al. (2024) is measured using **CLIPScore** and **BERTScore**, which are standard metrics for text–image and text–text alignment. CLIP Radford et al. (2021b) provides cross-modal contrastive alignment, while BERT Devlin et al. (2019) captures contextual semantics through bidirectional encoding.

Faithfulness is assessed using **concept insertion** and **concept deletion** curves Fel et al. (2023a); Kadir et al. (2023), which quantify performance shifts as important concepts are progressively inserted or removed. We report results across the top-1, top-2, and top-3 concepts ranked by their influence on model output.

Finally, qualitative analyses highlight representative extracted concepts and their textual groundings, and illustrate their application to binary classification and text-to-concept alignment.

### 5.3 RESULTS

Table 2 reports quantitative results for **concept vectors** extracted from **Gemma-3n-E4B** on ImageNet and MSCOCO. We compare CGDL with CoX-LMM across three dictionary learning settings (SNMF, SAE, SIMPLE). CGDL–SNMF consistently achieves the best performance, with the highest sparsity, lowest stability, and lowest overlap. SAE also benefits from concept guidance, while SIMPLE remains weak with low sparsity and high overlap. These results highlight that concept guidance substantially improves the quality of learned concept vectors, particularly under SNMF Fel et al. (2023a); Parekh et al. (2024).

| Method | Dictionary Learning | ImageNet | | | MSCOCO | | |
|--------|---------------------|-----------|----------|------------|-----------|----------|------------|
| | | Spars. ↑ | Stab. ↓ | Overlap ↓ | Spars. ↑ | Stab. ↓ | Overlap ↓ |
| CoX-LMM | SNMF | 0.96 | 0.13 | 0.25 | **1.00** | 0.02 | 0.16 |
| | SAE | 0.84 | 0.21 | 0.27 | 0.97 | 0.17 | 0.28 |
| | SIMPLE | 0.07 | 0.79 | 0.68 | 0.63 | 0.91 | 0.84 |
| CGDL | SNMF | **1.00** | **0.02** | **0.08** | **1.00** | **0.00** | **0.06** |
| | SAE | 0.90 | 0.16 | 0.10 | **1.00** | 0.05 | 0.16 |
| | SIMPLE | 0.52 | 0.64 | 0.67 | 0.80 | 0.49 | 0.54 |

Table 2: Concept vector evaluation on ImageNet and MSCOCO. Higher sparsity is better; lower stability and overlap are better. CGDL–SNMF yields the best results across both datasets.

Table 3 presents attribution results for **Gemma-3n-E4B** on ImageNet and MSCOCO. We evaluate two aspects of alignment: (i) **BERTScore**, which measures semantic correspondence between top-activated concept groundings and ground-truth labels, and (ii) **CLIPScore**, which assesses multimodal consistency between concept groundings and input images. We compare our method (CGDL) against CoX-LMM under three settings: **Text-only**, where activations are probed via class names; **Image-only**, where short visual descriptions are used; and **Combined**, where the model predicts concept presence ($c_k$ vs. UNK). Random baselines correspond to assigning concepts uniformly at random, which yields nearly constant values due to the data distribution. Across datasets and metrics, **CGDL with SNMF** consistently achieves stronger alignment and outperforms CoX-LMM in all modalities, despite requiring substantially less supervision. This advantage holds across both small-scale (10 concepts, MSCOCO) and large-scale (1,000 concepts, ImageNet) evaluations.

We evaluate the *concept attribution ranking* using established *faithfulness metrics* Fel et al. (2023a), namely **concept deletion (C-Deletion)** and **concept insertion (C-Insertion)**, on the MSCOCO validation set. For each image, the model is prompted to classify the input, and the top-3 activated concepts are identified from the residual embeddings (Sec. 4.4). Attribution faithfulness is then measured as follows:

1. **C-Deletion:** progressively set to zero the coordinates corresponding to the most influential concept directions, ranked by gradient magnitude with respect to the highest-probability token, and record the drop in output probability.
2. **C-Insertion:** start from a zero vector and gradually add concept coordinates in the same order, recording the corresponding probability increase.

Scores are averaged across tokens for each image, then aggregated over the validation set. Results are summarized in Figure 1.

Table 3: Comparison of CGDL and CoX-LMM on CLIPScore and BERTScore across datasets and concept types. Higher metric values indicate better alignment.

| Method | Dataset | Concept | Metric | Random | Text-only | Image-only | Combined |
|--------|---------|---------|--------|--------|-----------|------------|----------|
| CGDL | ImageNet | 1,000 | CLIPScore | 0.52 ± 0.04 | – | **0.62 ± 0.08** | **0.67 ± 0.09** |
| | | | BERTScore | 0.71 ± 0.06 | 0.78 ± 0.07 | 0.84 ± 0.08 | 0.86 ± 0.10 |
| | MSCOCO | 10 | CLIPScore | 0.48 ± 0.04 | – | 0.60 ± 0.06 | 0.64 ± 0.08 |
| | | | BERTScore | 0.71 ± 0.01 | **0.89 ± 0.06** | **0.91 ± 0.05** | **0.93 ± 0.07** |
| CoX-LMM | ImageNet | 1,000 | CLIPScore | 0.49 ± 0.04 | – | 0.57 ± 0.03 | 0.58 ± 0.05 |
| | | | BERTScore | 0.71 ± 0.00 | 0.74 ± 0.08 | 0.75 ± 0.06 | 0.82 ± 0.09 |
| | MSCOCO | 10 | CLIPScore | 0.51 ± 0.04 | – | 0.57 ± 0.10 | 0.55 ± 0.05 |
| | | | BERTScore | 0.71 ± 0.01 | 0.83 ± 0.01 | 0.79 ± 0.09 | 0.73 ± 0.11 |

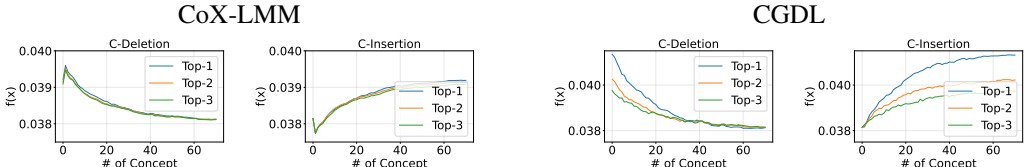

Figure 1: Faithfulness comparison of CoX-LMM and CGDL using concept deletion and insertion. CoX-LMM yields relatively flatter curves with weak separation across ranks, whereas CGDL preserves a clear order (Top-1 >Top-2 >Top-3), with sharper degradation under deletion and stronger recovery under insertion. This shows that CGDL produces more faithful and discriminative concept rankings.

Table 4: Mean ± std. of CLIPScore across binary prompts on MSCOCO-10. Abbreviations: Q-2 = Qwen-2, Q-2.5 = Qwen2.5, G-3n = Gemma-3n.

| Prompt | Q-2 | Q-2.5 | G-3n | Prompt | Q-2 | Q-2.5 | G-3n |
|--------|-----|-------|------|--------|-----|-------|------|
| P1 | 0.57±0.10 | 0.65±0.13 | 0.62±0.08 | P3 | 0.57±0.14 | 0.62±0.07 | 0.62±0.06 |
| P2 | 0.58±0.07 | 0.63±0.11 | 0.64±0.08 | P4 | 0.59±0.11 | 0.63±0.08 | 0.64±0.09 |

Figure 2 compares ImageNet concepts extracted by CoX-LMM (left) and CGDL (right). CGDL yields fine-grained, monosemantic representations (e.g., fur, stripes), whereas CoX-LMM produces entangled groundings that mix semantics (e.g., tiger conflated with lion or multiple animals). Figure 3 shows attribution in three settings: (i) binary classification, (ii) open-ended classification, and (iii) text–image alignment. In each case, attribution is explained by retrieving the nearest concept examples to the residual activation, demonstrating robust multimodal alignment.

## 6 ABLATION STUDY

We test multiple variants of the contrastive prompt template (see AppendixB for details). Table 4 shows only minor fluctuations in CLIPScore, indicating robustness to phrasing. Qwen-2 exhibits slightly higher variance than Qwen2.5 and Gemma-3n.

Furthermore, we ablate the SNMF sparsity weight $\alpha$ and dictionary sizes $K > 2$ and find that attribution results remain stable, while CLIP scores decrease for large $K$. Hence, we select $\alpha = 20$ and $K = 2$, which we find sufficient for attribution. Full results are reported in Appendix F. We also compare SAM with a simpler random-cropping baseline. SAM yields slightly higher BERTScore and CLIPScore, as its segmentation better localizes concepts in raw images. Detailed numerical results are reported in Appendix F, Table 10, and qualitative examples are shown in Figures 7 and 5.

## CONCLUSION

We introduced *Concept-Guided Dictionary Learning* (CGDL), a weakly supervised framework that enforces monosemanticity and grounds concepts directly within LVLMs, yielding faithful multimodal alignment. CGDL is flexible, efficient, and improves concept quality across dictionary learning

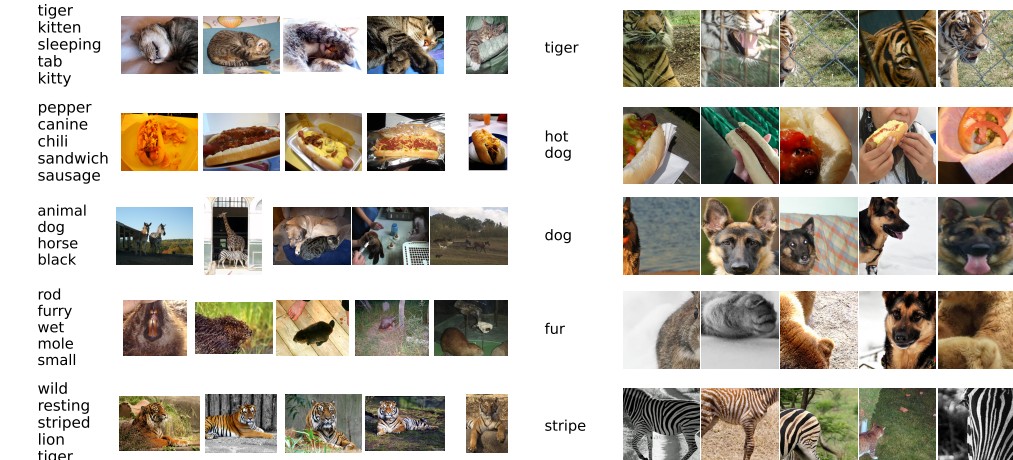

Figure 2: Qualitative examples of concept representations extracted from ImageNet. CoX-LMM (left) concepts are often grounded to multiple unrelated or overlapping tokens (e.g., "canine" linked to "hot dog"), reflecting polysemantic vectors. In several cases, concepts mix distinct animals: for example, *tiger* grounds across multiple concepts, while *lion* is misrepresented as *tiger*. In contrast, CGDL (right) discovers fine-grained and monosemantic concepts (e.g., fur, dog, stripes).

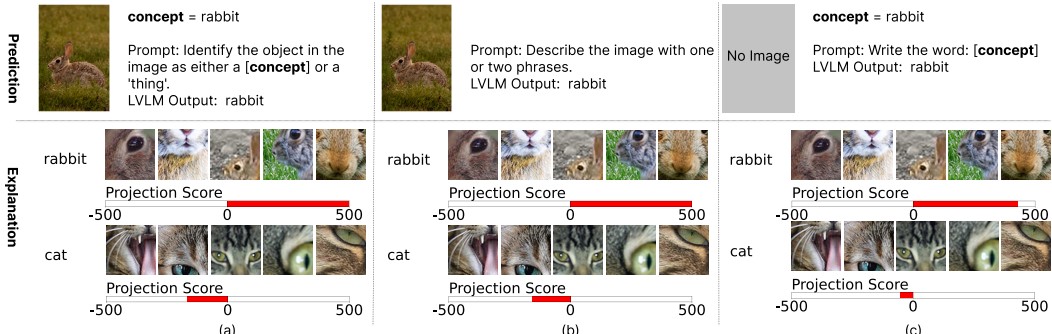

Figure 3: Attribution-based explanation using concepts (a) Aligned image-text concepts yield strong feature attribution. (b) An image-only input without concept information still triggers a relevant feature. (c) Text alone activates semantically meaningful features, demonstrating robust multimodal alignment. For token attribution with concepts from the same object, different objects, or abstract concepts, we provide more examples in Appendix E.3, E.5, and E.4. We find that our concepts can attribute a model's output both to abstract concepts and to objects from similar categories.

methods. Limitations include the lack of hierarchical organization and the requirement that models understand basic language instructions—though this holds for most modern LVLMs. Future work will extend CGDL to capture hierarchical concepts and to evaluate beyond LVLMs. CGDL relies on the target LVLM being able to follow prompts of similar difficulty to those in Appendix B, so its applicability is tied to the LVLM's prompt-understanding ability.

**Reproducibility Statement** We release a reproducible pipeline that requires only a Hugging Face model card, an access token, and a dataset directory with `train`/`val` splits. CGDL and CoX-LMM can be run via `scripts/run_full_pipeline.sh` and `scripts/run_full_pipeline_dl.sh`, respectively. Code and configs are shared anonymously at `https://anonymous.4open.science/r/xl-vlms-30C1`, with installation and usage detailed in the `README`. All experiments used a single NVIDIA RTX 3090 (24GB) GPU with fixed random seeds.

**Ethics Statement.** This work poses no direct risks beyond standard interpretability concerns. Our method may reveal biased concepts, which should be handled responsibly. We used an LLM for minor editing; all scientific contributions are our own.

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

**Appendix**

## A MONOSEMANTIC VS. POLYSEMANTIC REPRESENTATIONS

A central challenge in interpreting large vision–language models (LVLMs) lies in *superposition* and *feature entanglement* in high-dimensional residual streams (Elhage et al., 2022). Here, *features* can be understood as vector directions in activation space that encode candidate concepts. Ideally, such vectors should be *monosemantic*-each aligned with a single interpretable concept. In practice, however, LVLMs often learn *polysemantic* vectors, where multiple, semantically unrelated concepts activate a single direction.

For instance, in Fig. 4(a), a feature $f_1$ responds to both "cat" and "chair." Such overlap can arise when these concepts frequently co-occur in training data, leading the model to conflate them. When $f_1$ activated, it is therefore ambiguous whether the cause was the presence of a cat, a chair, or both. This ambiguity breaks the one-to-one mapping between features and concepts, making attribution unreliable. In this paper, we investigate how to extract monosemantic concept vectors from the polysemantic features that neurons of a model fire during prediction.

Monosemantic features (Fig. 4b) provide a clean relation to concepts: e.g., A concept vector is an approximation of monosemantic features.

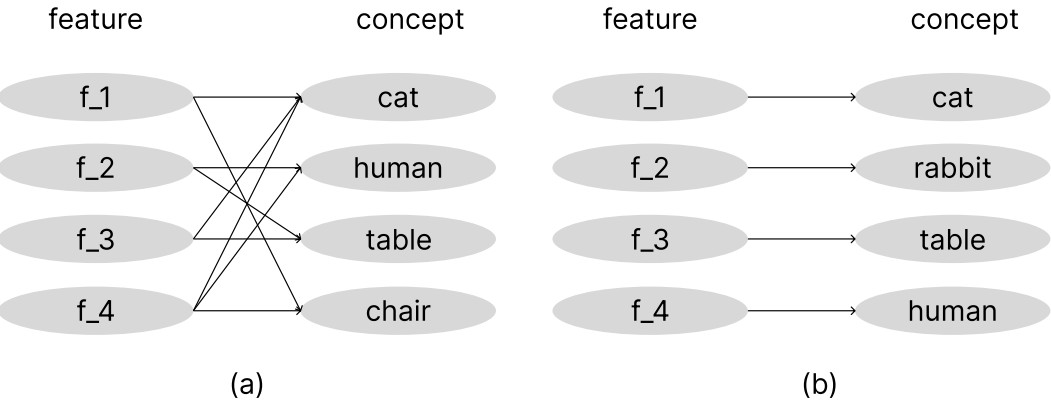

Figure 4: Comparison between prior concept decomposition methods and our proposed approach. (a) Previous methods (e.g., Parekh et al. (2024)) often produce polysemantic features (e.g., $f_3$ activates for "chair" and "cat"). (b) Our method encourages monosemantic features (e.g., $f_1$ for "cat," $f_3$ for "table").

Apart from a limited number of studies Templeton (2024); Pach et al. (2025), most existing models do not offer disentangled representations, and tools for analyzing and extracting monosemantic features remain scarce. This paper addresses this gap through **CGDL**.

## B PROMPTS

**Concept generation prompt.** We use the following instruction to extract candidate concept text from images: *"Identify every visible object, item, concept, and pattern in the image. Output only a single-word, comma-separated list. No explanations or sentences."* This prompt generates concept tokens directly from the dataset without requiring manual labels, thereby creating a concept-to-image mapping.

**Concept-Guidance prompt**

- P1: Detect whether the image contains $c_k$. If yes, return $c_k$; otherwise, return UNK.
- P2: Does the image contain $c_k$? If yes, output $c_k$; otherwise, output No-$c_k$.

- P3: Is there a clear instance of $c_k$ in this image? Reply with $c_k$ or `thing`, nothing else.
- P4: Recognize whether the concept $c_k$ is present in the picture. Use only $c_k$ or `UNK` as your answer.

## C  MODELS

### C.1  GEMMA-3N E4B-IT TEAM (2025A)

Gemma-3n E4B-IT (4-billion-parameter model) is trained using a nested subnetwork approach based on the Matryoshka Transformer (MatFormer) architecture. Each Transformer layer supports multiple capacity levels, implemented as top-left submatrices of full-size weight tensors. The model is instruction-tuned on a mixture of multilingual and multimodal data, including text, images, audio, and video inputs. It is trained autoregressively to predict the next token, with a maximum context length of 32k tokens.

### C.2  QWEN2.5-VL-7B-INSTRUCT TEAM (2025B) AND QWEN2-VL-7B-INSTRUCT WANG ET AL. (2024)

Qwen2.5-VL-7B-Instruct is a 7-billion-parameter multimodal instruction-tuned language model designed for vision-language tasks. It accepts both text and image inputs and generates text outputs. The model supports a maximum context length of 8192 tokens, enabling it to handle long conversational and reasoning scenarios. Training is performed via supervised instruction tuning on paired text-image datasets. The model is optimized with an autoregressive next-token prediction objective using cross-entropy loss, conditioning on both textual and visual contexts. Large-scale distributed training with mixed precision improves efficiency. This approach enhances the model's instruction-following capabilities and generalization to diverse vision-language tasks.

## D  DATASET DESCRIPTIONS

We evaluate our models on four datasets: **ImageNet**, **MSCOCO (10 classes)**, **CIFAR100**, and **DTD (Describable Textures Dataset)**.

- **ImageNet**: A large-scale visual dataset with 1000 object categories and high-resolution images, commonly used for image classification tasks.
- **MSCOCO (10 classes)**: A subset of the MSCOCO dataset with 10 object categories, featuring complex scenes and multiple annotated objects per image.

| Dataset | Total Classes | Train Samples/Class | Val Samples/Class |
|---------|---------------|---------------------|-------------------|
| ImageNet | 1000 | 300 | 50 |
| MSCOCO (10) | 10 | 300 | 50 |
| CIFAR100 | 100 | 300 | 100 |
| DTD | 47 | 95 | 24 |

Table 5: Overview of datasets used for training and validation, including the number of classes and samples per class.

## E  RESULTS

### E.1  QWEN 2.0-VL-7B

Table 6 reports the performance of **CGDL** and **CoX-LMM** on four benchmark datasets. We evaluate alignment using **CLIPScore (CS)** and **BERTScore (BS)**, where higher is better.

Across all datasets, CGDL consistently outperforms CoX-LMM. Notably, on **MSCOCO**, CGDL achieves the strongest gains: BS improves from 0.82 (CoX-LMM) to **0.94**, and CS improves from

Table 6: Comparison of **CGDL** and **CoX-LMM** across datasets. Metrics: CS = CLIPScore, BS = BERTScore. Higher is better. The best non-random results are **bolded**.

| Method | Dataset | #C | Metric | Rand | Text | Img | Comb |
|--------|---------|-----|--------|------|------|-----|------|
| CGDL | ImageNet | 1k | CS | 0.53 ± 0.02 | – | **0.62 ± 0.07** | **0.63 ± 0.08** |
| | | | BS | 0.80 ± 0.05 | **0.86 ± 0.06** | **0.88 ± 0.08** | **0.86 ± 0.09** |
| | CIFAR100 | 100 | CS | 0.51 ± 0.02 | – | **0.63 ± 0.04** | **0.63 ± 0.05** |
| | | | BS | 0.83 ± 0.08 | **0.86 ± 0.07** | **0.87 ± 0.08** | **0.91 ± 0.08** |
| | DTD | 47 | CS | 0.53 ± 0.06 | – | **0.63 ± 0.05** | **0.62 ± 0.05** |
| | | | BS | 0.74 ± 0.04 | **0.84 ± 0.06** | **0.83 ± 0.07** | **0.87 ± 0.06** |
| | MSCOCO | 10 | CS | 0.52 ± 0.03 | – | **0.64 ± 0.06** | **0.62 ± 0.06** |
| | | | BS | 0.82 ± 0.02 | **0.88 ± 0.06** | **0.89 ± 0.05** | **0.94 ± 0.08** |
| CoX-LMM | ImageNet | 1k | CS | 0.53 ± 0.03 | – | 0.54 ± 0.03 | 0.53 ± 0.05 |
| | | | BS | 0.82 ± 0.04 | 0.82 ± 0.03 | 0.84 ± 0.04 | 0.86 ± 0.09 |
| | CIFAR100 | 100 | CS | 0.53 ± 0.04 | – | 0.53 ± 0.06 | 0.53 ± 0.05 |
| | | | BS | 0.78 ± 0.06 | 0.84 ± 0.06 | 0.80 ± 0.03 | 0.73 ± 0.07 |
| | DTD | 47 | CS | 0.51 ± 0.05 | – | 0.54 ± 0.05 | 0.52 ± 0.05 |
| | | | BS | 0.80 ± 0.07 | 0.84 ± 0.06 | 0.83 ± 0.07 | 0.77 ± 0.07 |
| | MSCOCO | 10 | CS | 0.53 ± 0.03 | – | 0.57 ± 0.04 | 0.58 ± 0.06 |
| | | | BS | 0.82 ± 0.04 | 0.83 ± 0.01 | 0.83 ± 0.05 | 0.82 ± 0.03 |

0.58 to **0.64**. Similarly, on **CIFAR100**, the BS of CoX-LMM drops to 0.73 in the combined setting, while CGDL achieves a significantly higher **0.91**. These results highlight the robustness of CGDL in both low- and high-concept regimes.

### E.2 QWEN 2.5-VL-7B

Table 7 shows results for the Qwen2.5 backbone. Again, CGDL achieves the strongest improvements across all datasets. In particular, on **MSCOCO**, CGDL improves BS from 0.90 to **0.95** in the combined setting. On **CIFAR100**, CGDL reaches **0.92**, compared to 0.87 with CoX-LMM. These consistent gains indicate that CGDL scales effectively from small (10 concepts) to large-scale (1k concepts) benchmarks.

### E.3 POSTHOC CONCEPT EXPLANATIONS FOR LVLMS

Unlike the existing methods, which can't provide any token-level posthoc explanation, our method provides token-level explanations in autoregressive Large Vision-Language Models (LVLMs). We present qualitative examples in Figure 7 5 and 8 for Qwen2.5-VL-7B. Figure 5 results belong to the experiemnt wsing SAM as a localizer mentioned in 4.1 , while Figure 7 and 8 present the concept attrribution example using randdom cropping localizer during concept. Each example uses a structured $2 \times 2$ image grid that intentionally makes the prediction task more challenging while encouraging the model to produce structured, multi-object descriptions.

On the **left** of each example, we show the *input image* (the $2 \times 2$ grid) together with the *prompt* used for generation. Directly below, we display the *LVLM's textual output* produced for this visual input.

On the **right**, we visualize token-wise concept activations. For each generated token, we extract its penultimate-layer embedding and compute its cosine similarity to all learned concept vectors. For token-to-word mapping, we consider only the first produced token embedding for that position when calculating the cosine distance to the concept vectors. We then identify the top-2 most activated concepts (from left to right) corresponding to that token.

The right-hand panel contains:

- a *concept grid* showing, for each token, the two highest-scoring concepts;

| Method | Dataset | #C | Metric | Rand | Text | Img | Comb |
|--------|---------|-----|--------|------|------|-----|------|
| CGDL | ImageNet | 1k | CS | 0.51 ± 0.06 | – | **0.63 ± 0.06** | **0.64 ± 0.05** |
| | | | BS | 0.82 ± 0.04 | **0.87 ± 0.01** | **0.88 ± 0.07** | **0.87 ± 0.07** |
| | MSCOCO | 10 | CS | 0.53 ± 0.05 | – | **0.64 ± 0.07** | **0.64 ± 0.08** |
| | | | BS | 0.83 ± 0.06 | **0.89 ± 0.07** | **0.90 ± 0.06** | **0.95 ± 0.08** |
| | CIFAR100 | 100 | CS | 0.50 ± 0.07 | – | **0.62 ± 0.05** | **0.64 ± 0.06** |
| | | | BS | 0.80 ± 0.08 | **0.88 ± 0.06** | **0.88 ± 0.09** | **0.92 ± 0.07** |
| | DTD | 47 | CS | 0.51 ± 0.06 | – | **0.63 ± 0.08** | **0.63 ± 0.07** |
| | | | BS | 0.79 ± 0.07 | **0.87 ± 0.07** | **0.85 ± 0.09** | **0.89 ± 0.08** |
| CoX-LMM | ImageNet | 1k | CS | 0.51 ± 0.05 | – | 0.56 ± 0.04 | 0.55 ± 0.06 |
| | | | BS | 0.81 ± 0.06 | 0.83 ± 0.04 | 0.85 ± 0.06 | 0.86 ± 0.03 |
| | CIFAR100 | 100 | CS | 0.53 ± 0.07 | – | 0.58 ± 0.06 | 0.57 ± 0.04 |
| | | | BS | 0.78 ± 0.07 | 0.84 ± 0.06 | 0.85 ± 0.05 | 0.87 ± 0.06 |
| | MSCOCO | 10 | CS | 0.52 ± 0.03 | – | 0.60 ± 0.03 | 0.60 ± 0.02 |
| | | | BS | 0.81 ± 0.04 | 0.88 ± 0.04 | 0.90 ± 0.04 | 0.90 ± 0.07 |
| | DTD | 47 | CS | 0.53 ± 0.03 | – | 0.56 ± 0.03 | 0.56 ± 0.06 |
| | | | BS | 0.81 ± 0.04 | 0.83 ± 0.05 | 0.82 ± 0.06 | 0.88 ± 0.05 |

Table 7: Comparison of CGDL and CoX-LMM across datasets using CLIPScore (CS) and BERTScore (BS). Best non-random results are **bolded**.

- a *concept bank* in the form of a row of thumbnails, starts with a similarity bar where it shows the cosine similarity between the token embedding and the corresponding concept vector;

- a short *textual grounding label* above each concept bank, summarizing the semantic meaning of the discovered concept.

We find that SAM and the random localizer perform similarly in attribution ranking: both methods correctly attribute the concept images. The main difference is that SAM yields better visualizations for concepts such as hot-dog, beaver, and bear in the concept bank. Moreover, in Figure 6, we show qualitative examples of explanations produced by the baseline CoX-LMM to highlight its limitations and how our method improves the ranking of similar concepts. While the CoX-LMM concept vector often shows high cosine similarity with the token activations, the corresponding concept bank contains many different object types. This indicates that multiple concept signals are entangled in a single vector, i.e., the concept vectors are polysemic. Such explanations are less useful because we cannot reliably map the model output to a specific, human-interpretable concept. In addition, some textual concepts associated with the concept bank are not clearly related to the underlying visual patterns.

### E.4 CONCEPT VECTORS ARE GENERALIZABLE TO RELATED OBJECTS

To study the robustness and generalization of our concept-based explanations, we also analyze cases where the *object-specific concept is missing in the concept dictionary*. In Figure 9, we provide qualitative examples showing how the LVLM aligns its predictions with the *closest available* concepts, even when the extracted concept set belongs to different object classes than the input images.

As in the previous section, E.3, each example contains a structured $2 \times 2$ input grid. On the left side of each example, we show the input image, the prompt, and the LVLM's generated output. Here, the learned concepts originate from *different object categories* than the input images. Despite this mismatch, the LVLM often activates semantically *related* concepts whose attributes partially overlap with the visual content (e.g., "stripe-like patterns", " texture", "fur-like appearance").

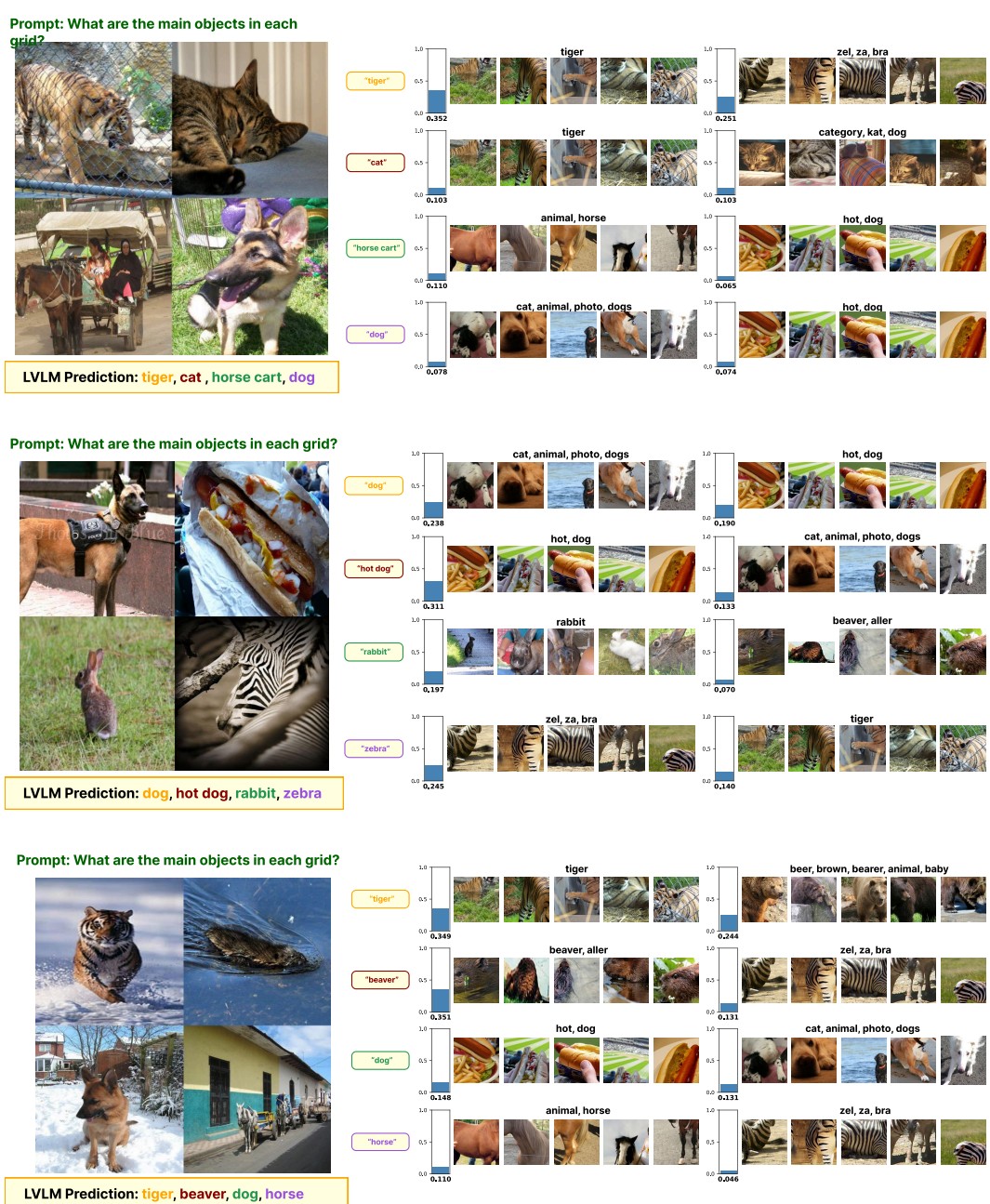

Figure 5: **Left: input $2 \times 2$ grid, prompt, and LVLM output. Right: token-wise top-2 concept activations, cosine-similarity bars, and textual grounding and visual grounding. Example using SAM as an object localizer during concept extraction.**

### E.5 EXPLANATION WITH ABSTRACT CONCEPTS

We explored how abstract concepts relate to an LVLM's outputs using ImageNet validation examples. In Figure 10, we present three cases. We found that our explanation method captures clear relationships between predicted tokens and related abstract concepts. For instance, *macaw* shows high similarity to *colorful* concepts; *ladybug* and *spotted dog* (Dalmatian) show high similarity to *polka-dot*; and *jellyfish* shows high similarity to *skin* and *soft* concepts. These results shed light on how the model may internally perform abstract reasoning when predicting a token.

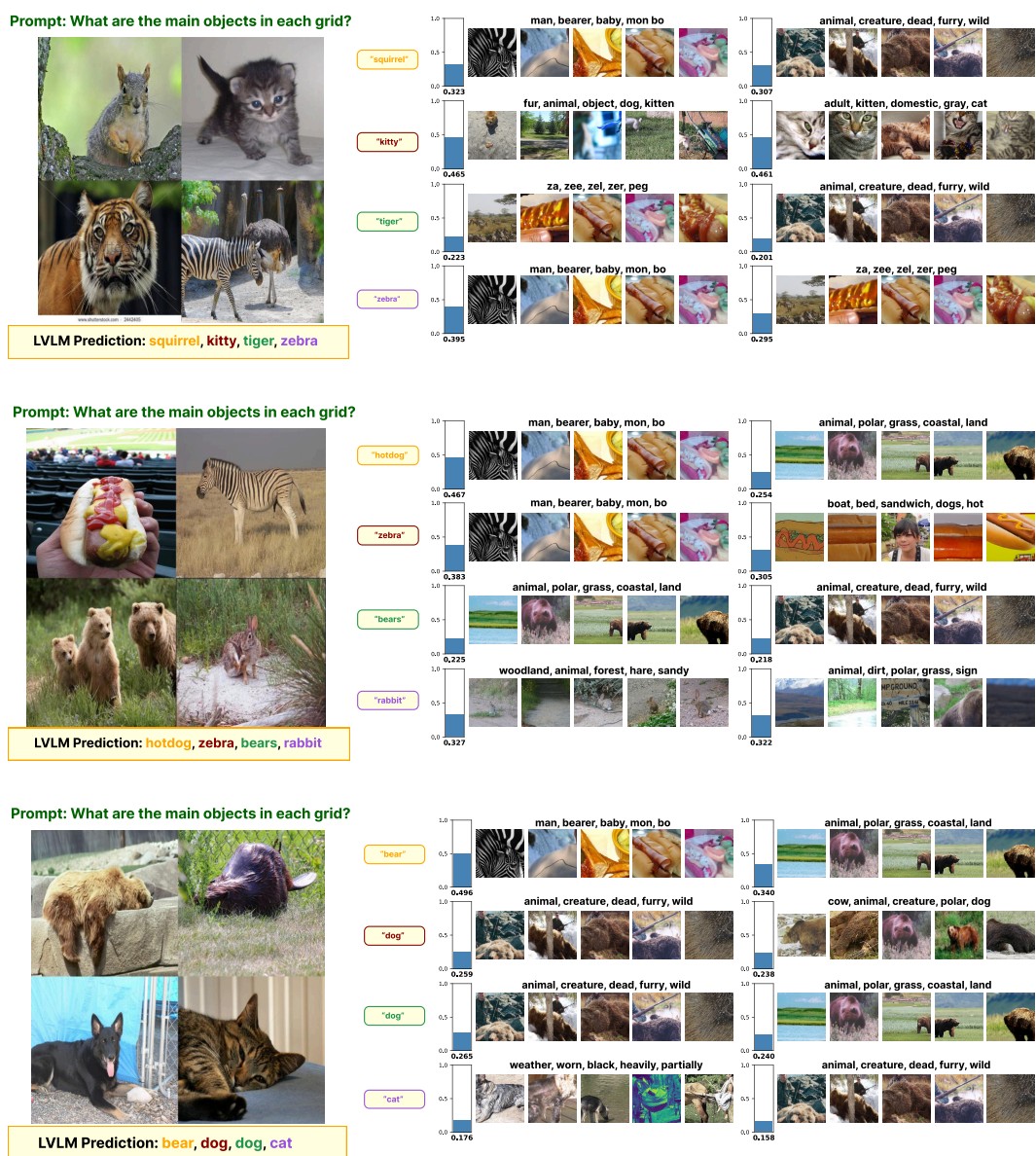

Figure 6: Left: input $2 \times 2$ image grid, prompt, and LVLM output. Right: token-wise top-2 concept activations, cosine-similarity bars, and textual/visual grounding obtained with the baseline CoX-LMM. SAM is used as an object localizer during concept extraction.

### E.6 GROUNDING LIMITATION

Although our concept-based attribution method generally provides coherent visual and textual grounding, we observe an important failure case when analyzing images containing a *beaver* property. Figure 11 illustrates this phenomenon.

**Shifted textual grounding in some examples.** *Textual grounding* is sometimes slightly shifted (Figure 11) or offset from the intended semantic meaning (e.g., the "cat" concept is grounded as "dog," "category," and "kat," and for "beaver," it shifted to the tokens "based" and "prediction"), while the visual grounding is consistent with the image of a "cat" and "beaver."

This discrepancy suggests that while the concept vectors are visually stable and reliably activated across different images, the mapping from concept vectors to text tokens remains sensitive to local

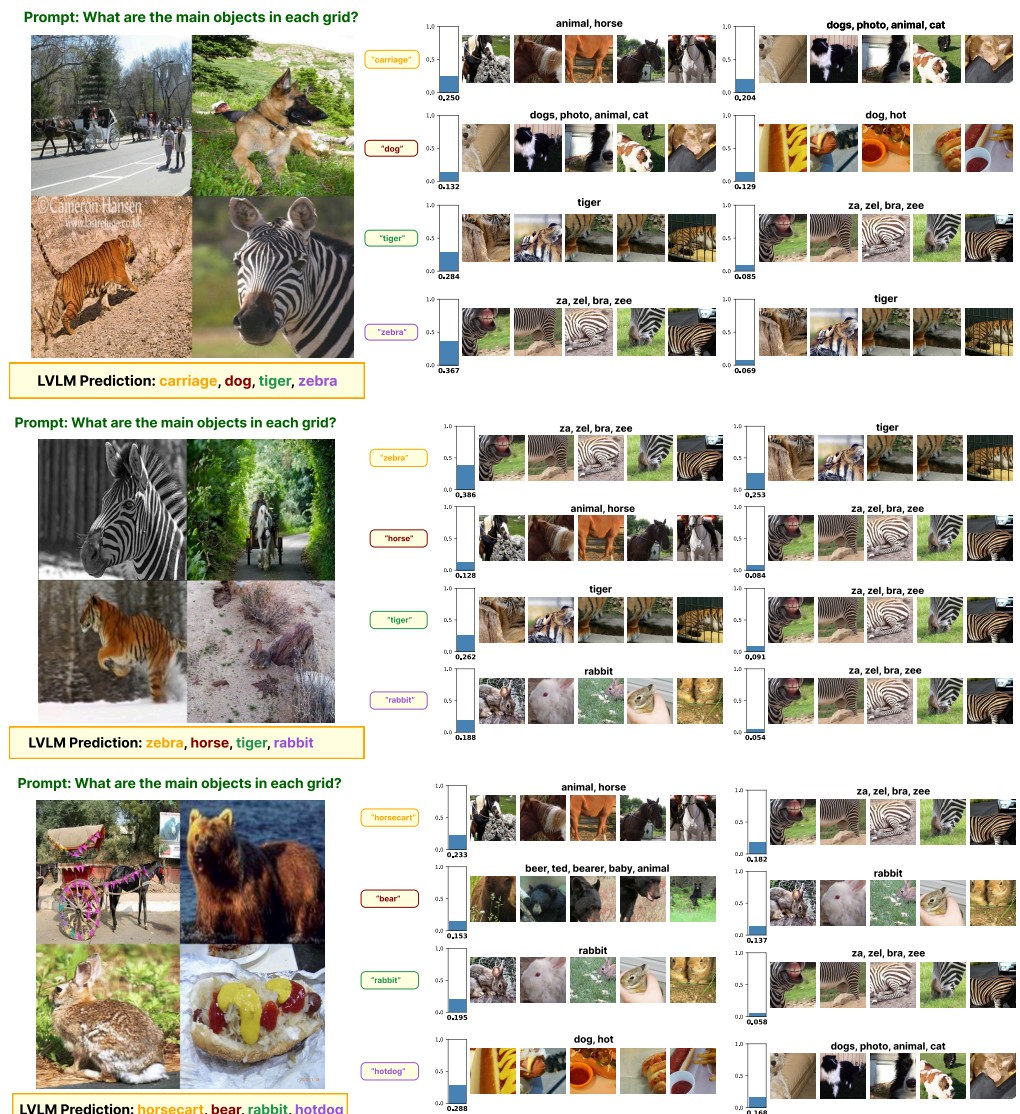

Figure 7: Example using random cropping as an object localizer during concept extraction.

variations in the LVLM's decoder distribution. As a result, text grounding may deviate slightly even when image grounding is fully correct. Overall, however, the image-side concept activations in our concept-based examples remain remarkably uniform and consistent across all inputs.

# F ABLATION STUDY

**Dictionary size ablation on Gemma-3n.** We perform an ablation on the dictionary size $K$ (number of atoms) (Table 8) using Gemma-3n. Dictionaries are learned on ImageNet training data and evaluated on the ImageNet validation split from the same five object classes (chosen to reduce computation). For each $K$, we measure CLIPScore and BERT scores for image–text alignment and BERTScore for semantic similarity of the concept phrases.

**SNMF $\alpha$ ablation on Gemma-3n.** We also ablate the SNMF sparsity weight $\alpha$ while keeping the dictionary size fixed. Larger $\alpha$ promotes sparser and more selective atoms, while smaller $\alpha$ yields denser activations. We train on ImageNet training images from the same five classes and evaluate on the corresponding validation split, reporting CLIPScore and BERTScore for each $\alpha$. As shown in

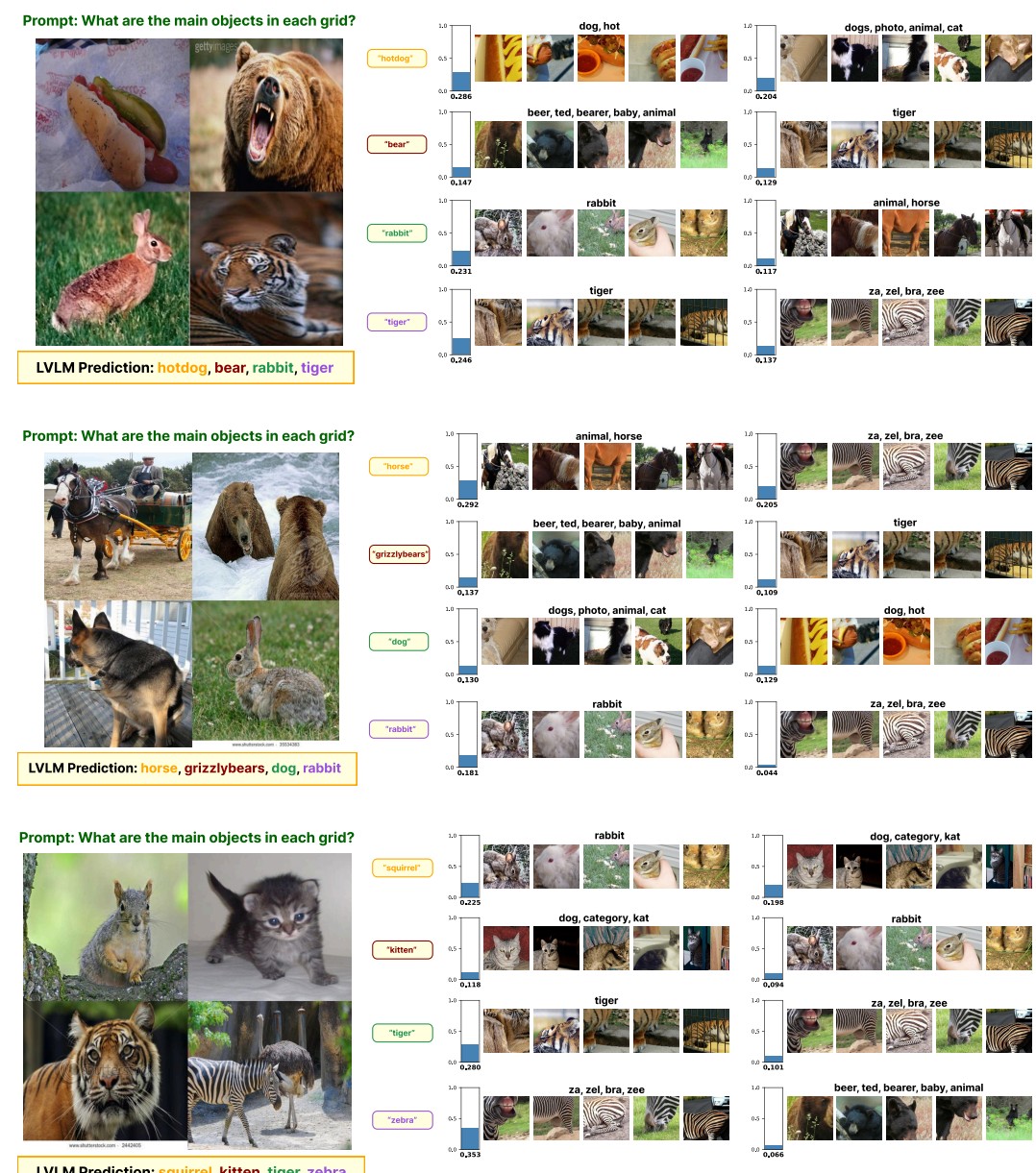

Figure 8: **More examples using random localizer**

Table 9, increasing $\alpha$ gives only a slight improvement and the overall differences are small. Based on this study, we use $\alpha = 20$ in all main experiments.

**SAM vs. non-SAM localization ablation.** We further ablate the image localization step by comparing SAM-based region proposals with a non-SAM baseline (random/local crops), while keeping the LVLM (Gemma-3n) and dictionary settings fixed. Both variants are trained on ImageNet training images and evaluated on the validation split of the same five classes. We report CLIPScore and BERTScore to quantify concept quality. As shown in Fig. **??**, the two approaches achieve similar quantitative performance, but SAM produces cleaner and more visually coherent concept exemplars for some classes, improving qualitative interpretability.

**Layer Ablation** We extract concept vectors from different normalization layers of Gemma-3n and report their BERTScore and CLIPScore in Figure 12. We observe that the CLIP scores follow the

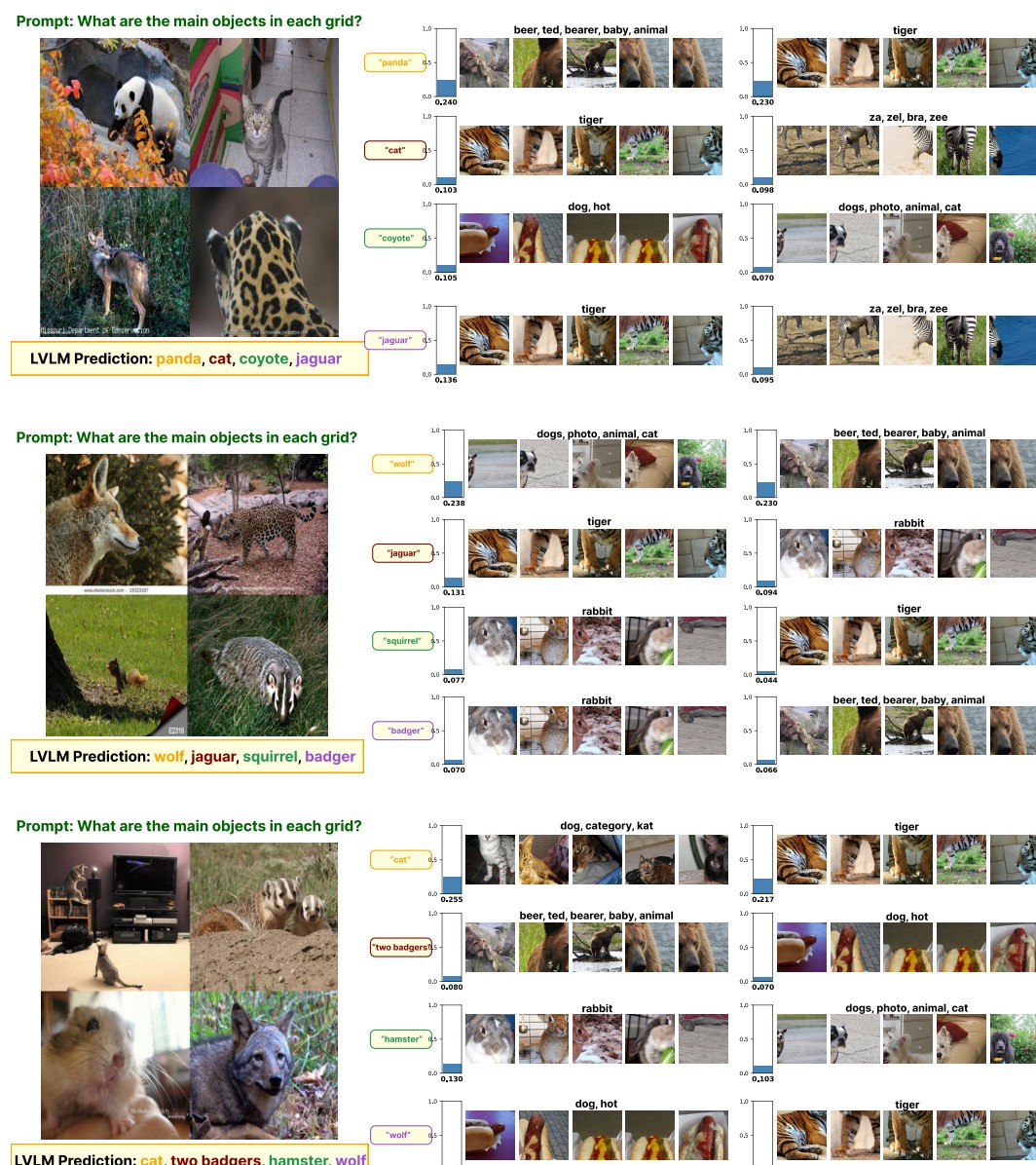

Figure 9: **Concept-Mismatch Analysis.** When the exact object's concept does not exist in the dictionary, the LVLM aligns its token embeddings with the most semantically related available concepts.

same trend as in prior work: they are generally higher for deeper layers (Parekh et al., 2024). This is expected, since deeper layers contain more global image features, and CLIPScore primarily measures global image–text similarity rather than fine-grained local details.

In contrast, the BERT scores do not increase monotonically with depth. Instead, they are high for some layers and low for others. This is reasonable because BERTScore only compares the text descriptions of the concepts. Text is discrete and does not decompose into "low-level" vs. "high-level" visual features in the same way as image representations, so BERTScore is largely agnostic to layer depth. A high BERTScore for a layer indicates that its concept vectors yield coherent and semantically rich textual concepts, whereas a low BERTScore suggests that the corresponding layer provides a poorer representation of the underlying concepts.

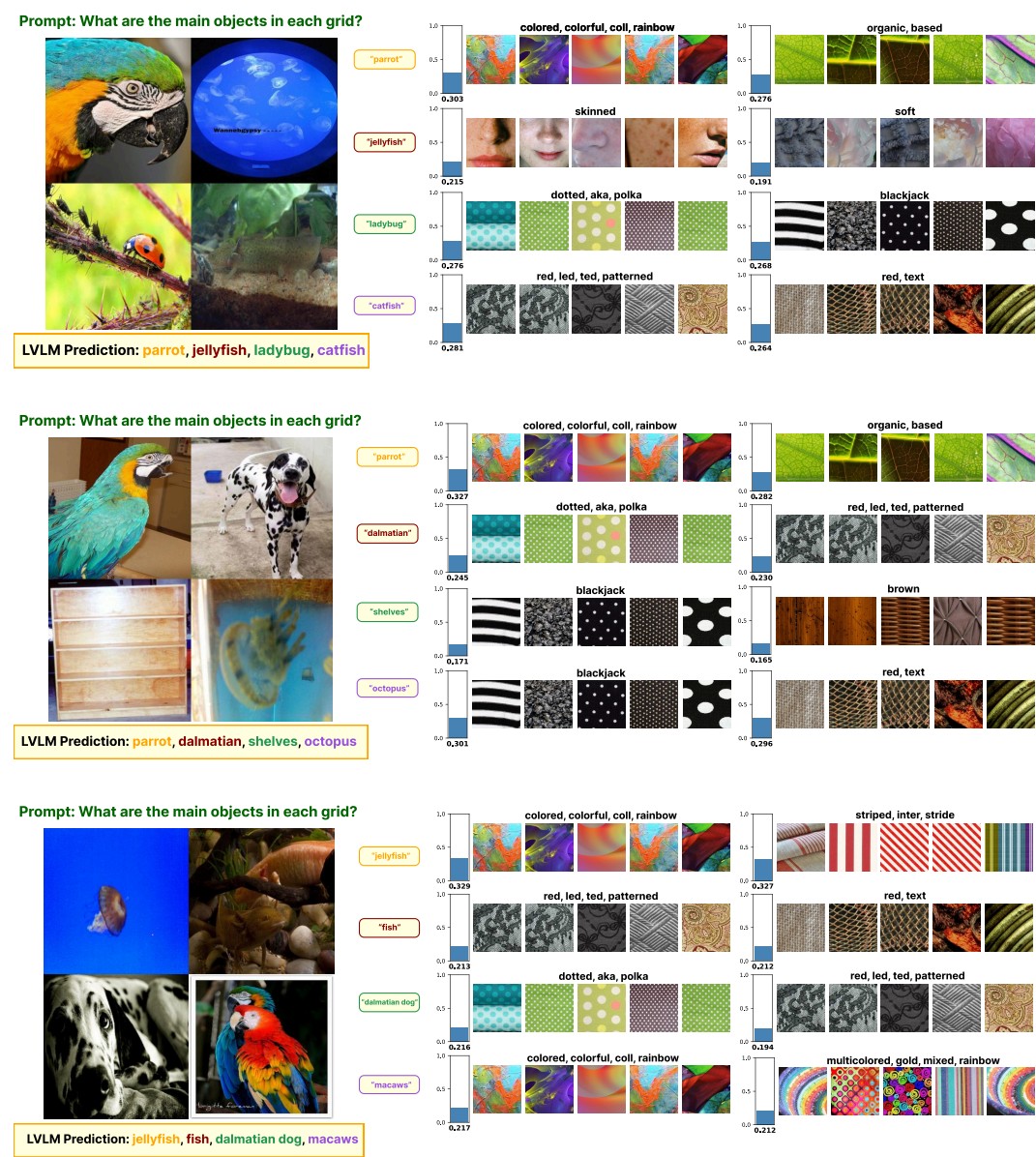

Figure 10: **Examples:** Top-2 concept activations reveal partial semantic similarity between token prediction and related abstract concepts.

## G  COMPUTATIONAL COST ANALYSIS

**Setup.** We compare the computational cost of **CGDL** and the **CoX-LMM** baseline under the same hardware: GPU: **NVIDIA RTX 3090 (24GB)**, CPU: **AMD Ryzen 9 5950X (16 cores)**. On this setup, extracting concepts for a fixed set of 10 objects takes **421 s** for CGDL and **4375 s** for CoX-LMM, assuming that the image–concept assignment (concept bag) is already available on the GPU.

**Summary.** Table 11 reports the estimated training and inference cost. CGDL requires substantially fewer FLOPs than CoX-LMM during concept learning ($\sim 40\%$ reduction in GPU FLOPs and $\sim 4\times$ fewer CPU operations), while the per-image inference cost is identical between the two methods.

**Decomposition of CGDL FLOPs.** We approximate the LVLM cost using a Gemma-3n-4B backbone with $M = 4 \times 10^9$ parameters and assume a per-token cost of $\approx 2M \approx 8 \times 10^9$ FLOPs.

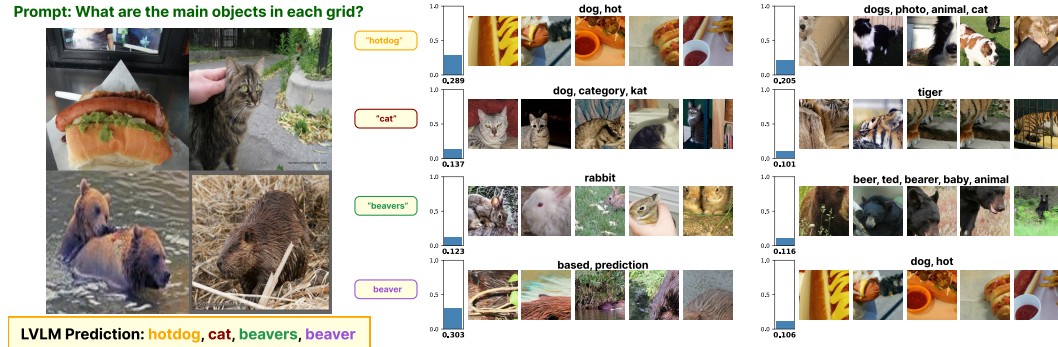

Figure 11: **Failure Case: LVLM Hallucination, Text Grounding Incorrect of CGDL**. LVLM predicts "bears'" image as beavers. The concept explanation relates more to a rabbit than a bear. This means that the model knows bears, but it just hallucinated due to the beaver-like appearance. Secondly, even though the textual grounding for "beaver" is incorrect in the last concept bank in the middle, image grounding is correct, and activation correctly responds to the beaver-like region (body on water). The textual grounding of the concept is incorrectly shifted due to interference (possibly "water-based"), demonstrating a misalignment between visual and textual grounding.

| $K$ | BERT@1 ↑ | BERT@2 ↑ | BERT@3 ↑ | CLIP@1 ↑ | CLIP@2 ↑ | CLIP@3 ↑ |
|---|---|---|---|---|---|---|
| 2 | 0.881±0.013 | 0.884±0.014 | 0.885±0.014 | 0.616±0.044 | 0.638±0.041 | 0.652±0.043 |
| 10 | 0.881±0.013 | 0.884±0.013 | 0.885±0.013 | 0.603±0.047 | 0.629±0.047 | 0.643±0.048 |
| 30 | 0.884±0.040 | 0.891±0.041 | 0.892±0.041 | 0.475±0.033 | 0.513±0.026 | 0.532±0.026 |
| 50 | 0.885±0.040 | 0.891±0.041 | 0.892±0.041 | 0.509±0.028 | 0.532±0.028 | 0.552±0.030 |
| 100 | 0.887±0.041 | 0.891±0.041 | 0.892±0.041 | 0.488±0.035 | 0.512±0.031 | 0.525±0.032 |

Table 8: Ablation over the number of concept atoms $K$ in the dictionary (five sampled settings). BERTScore stays roughly constant around 0.88, while CLIPScore generally decreases as $K$ increases. Values are mean±std over five ImageNet classes.

| $\alpha$ | BERTScore ↑ | | | CLIPScore ↑ | | |
|---|---|---|---|---|---|---|
| | @1 | @2 | @3 | @1 | @2 | @3 |
| 0 | 0.881 ± 0.013 | 0.884 ± 0.013 | 0.885 ± 0.013 | 0.610 ± 0.039 | 0.634 ± 0.036 | 0.646 ± 0.037 |
| 20 | 0.881 ± 0.013 | 0.884 ± 0.014 | 0.885 ± 0.014 | 0.616 ± 0.044 | 0.638 ± 0.041 | 0.652 ± 0.043 |
| 100 | 0.881 ± 0.013 | 0.884 ± 0.014 | 0.885 ± 0.014 | 0.615 ± 0.043 | 0.638 ± 0.040 | 0.652 ± 0.042 |
| 150 | 0.881 ± 0.013 | 0.884 ± 0.014 | 0.885 ± 0.013 | 0.617 ± 0.043 | 0.639 ± 0.038 | 0.653 ± 0.039 |
| 200 | 0.881 ± 0.013 | 0.883 ± 0.013 | 0.884 ± 0.014 | 0.621 ± 0.039 | 0.644 ± 0.039 | 0.659 ± 0.042 |

Table 9: SNMF sparsity weight $\alpha$ ablation on Gemma-3n (5 ImageNet classes). Values are mean ± std.

Let $T$ denote the total token length of the multimodal sequence (image tokens, prompt tokens, and generated tokens).

(1) CONCEPT-BAG CREATION (SEC. 4.1). We create concept bags from $N_{\text{img}} = 3000$ images (300 images per category) with token length $T = 196 + 40 + 200 = 436$ per sample. The total cost for this stage is

$$\text{FLOPs}_{\text{imgs}} \approx (8 \times 10^9) \cdot T \cdot N_{\text{img}} \approx 1.05 \times 10^{16}.$$

(2) SEGMENTATION COST (SAM) (SEC. 4.1). From these images we obtain 16,000 object-centric crops using SAM. Approximating the SAM forward cost as $\approx 1.1 \times 10^{12}$ FLOPs per image, we obtain

$$\text{FLOPs}_{\text{SAM}} \approx (1.1 \times 10^{12}) \cdot 3000 \approx 3.3 \times 10^{15}.$$

| Model | SAM | | Random | |
|---|---|---|---|---|
| | BERTScore@1 ↑ | CLIPScore@1 ↑ | BERTScore@1 ↑ | CLIPScore@1 ↑ |
| Gemma-3n | $0.93 \pm 0.04$ | $0.61 \pm 0.02$ | $0.88 \pm 0.01$ | $0.59 \pm 0.04$ |
| Qwen-2.5 | $0.92 \pm 0.03$ | $0.67 \pm 0.03$ | $0.91 \pm 0.06$ | $0.66 \pm 0.03$ |
| Qwen-2.0 | $0.93 \pm 0.04$ | $0.63 \pm 0.04$ | $0.91 \pm 0.06$ | $0.61 \pm 0.04$ |

Table 10: SAM vs. Random only localization ablation across three LVLM backbones. Values are mean $\pm$ std over five ImageNet classes. We notice that SAM improves both CLIP and BERT compared to random cropping for localization.

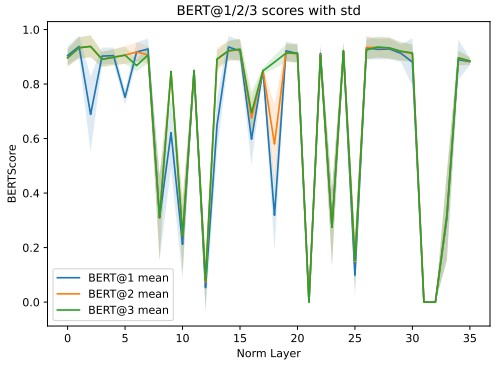

(a) BERTScore across norm layers.

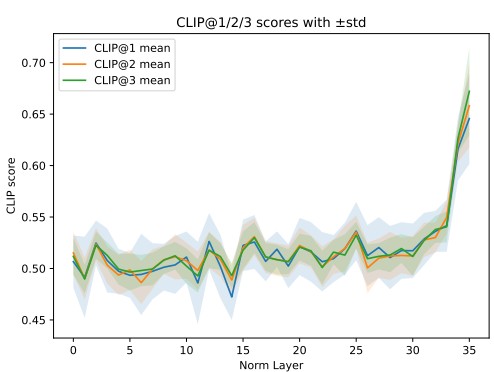

(b) CLIPScore across norm layers.

Figure 12: Layer-wise ablation on Gemma-3n. (a) BERTScore of concept phrases extracted from different norm layers. (b) CLIPScore between visual concepts and their text descriptions.

(3) RESIDUAL-STREAM EXTRACTION FOR CONCEPT BAGS (SEC. 4.2). For dictionary learning we use $N_{\text{res}} = 16{,}000$ residual samples (1600 crops per concept, 10 concepts). During binary prompting, the token length is $T_{\text{res}} \approx 196 + 21 + 10 = 227$. The LVLM FLOPs for residual extraction are

$$\text{FLOPs}_{\text{res}} \approx (8 \times 10^9) \cdot T_{\text{res}} \cdot N_{\text{res}} \approx 2.91 \times 10^{16}.$$

(4) SNMF DICTIONARY LEARNING ON CPU (SEC. 4.2). We factorize the residual activations with sparse NMF using $N = 1600$ samples, feature dimension $D = 2048$, dictionary size $K = 2$, 5000 iterations, and 10 concepts:

$$\text{Ops}_{\text{CGDL,SNMF}} \approx N \cdot D \cdot K \cdot \text{iters} \cdot \text{Concepts} = 1600 \cdot 2048 \cdot 2 \cdot 5000 \cdot 10 \approx 3.28 \times 10^{11} \text{ CPU ops.}$$

**Total CGDL cost.** Summing the LVLM FLOPs across stages yields

$$\text{FLOPs}_{\text{CGDL,total}} \approx 1.05 \times 10^{16} + 3.3 \times 10^{15} + 2.91 \times 10^{16} \approx 4.28 \times 10^{16}.$$

**Decomposition of CoX-LMM FLOPs.**

(1) CAPTION-LEVEL SEARCH (CPU). CoX-LMM first searches the MSCOCO captions to find images containing the target objects. This involves scanning $\approx 120{,}000$ captions, each of length $\approx 400$ characters, which leads to

$$\text{Ops}_{\text{CoX,search}} \approx 120{,}000 \times 400 = 4.8 \times 10^7 \text{ character comparisons.}$$

This cost is negligible compared to the LVLM forward passes.

(2) RESIDUAL-STREAM EXTRACTION. CoX-LMM extracts residual activations from 35,000 images ($\approx 3500$ images per object for 10 objects). The token length is $T_{\text{res}} \approx 196 + 50 + 11 = 257$, leading to

$$\text{FLOPs}_{\text{CoX,res}} \approx (8 \times 10^9) \cdot T_{\text{res}} \cdot 35{,}000 \approx 7.20 \times 10^{16}.$$

| Category | CGDL | CoX-LMM | Comment |
|---|---|---|---|
| Training GPU FLOPs | $\approx 4.28 \times 10^{16}$ | $\approx 7.20 \times 10^{16}$ | CGDL $\approx 40\%$ cheaper |
| Training CPU ops (dictionary learning) | $\approx 3.28 \times 10^{11}$ | $\approx 1.43 \times 10^{11}$ | CoX-LMM $\approx 4\times$ higher |
| Inference FLOPs / image | $\approx 1.6 \times 10^{11}$ | $\approx 1.6 \times 10^{11}$ | Cosine cost negligible |

Table 11: Training and inference cost of CGDL vs. CoX-LMM.

(3) DICTIONARY LEARNING ON CPU. CoX-LMM uses SNMF with $N = 35{,}000$ samples, $D = 2048$, $K = 10$ concepts, and 200 iterations:

$$\text{Ops}_{\text{CoX,SNMF}} \approx N \cdot D \cdot K \cdot \text{iters} = 35{,}000 \cdot 2048 \cdot 10 \cdot 200 \approx 1.43 \times 10^{11} \text{ CPU ops.}$$

**Total CoX-LMM cost.** The dominant cost for CoX-LMM is the LVLM residual extraction:

$$\text{FLOPs}_{\text{CoX,total}} \approx 7.20 \times 10^{16}.$$

The additional CPU cost from caption search and SNMF is small compared to the GPU FLOPs.

**Inference-time cost.** At inference time, both CGDL and CoX-LMM reuse the same LVLM backbone. For a single image, we approximate the LVLM cost as $\approx 1.6 \times 10^{11}$ FLOPs, and the extra cosine similarity operations between residual activations and concept vectors are negligible. Therefore, the per-image inference cost is effectively identical for both methods.

