# CONCEPT-GUIDED DICTIONARY LEARNING FOR INTERPRETABLE CONCEPT EXTRACTION AND ATTRIBUTION IN LARGE VISION–LANGUAGE MODELS

## ABSTRACT

Autoregressive Large Vision-Language Models (LVLMs) generate text sequentially, conditioning each token on evolving multimodal states. This makes it difficult to assess whether predictions are grounded in **visual concepts** or instead reflect hallucination or bias. Existing concept-discovery approaches such as **TCAV**, **CRAFT**, and **CLIP-Dissect** are designed for encoder-only or contrastive models. At the same time, recent LVLM methods (CoX-LMM) depend on labeled concepts and simplified settings, limiting scalability.

We propose **Concept-Guided Dictionary Learning (CGDL)**, a weakly supervised and scalable framework for multimodal concept discovery in autoregressive LVLMs. CGDL first probes the model to surface textual concepts from

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

Autoregressive Large Vision–Language Models (LVLMs) present new challenges: activations evolve across time steps, residual streams encode multiple dependencies, and concepts rarely align with a single hidden state (Templeton, 2024). CoX-LMM (Parekh et al., 2024) adapted Semi-NMF Trigeorgis et al. (2014) to LVLM activations, but struggles with (i) reliance on tokenized object names, (ii) not being suitable for multi-token concepts, (iii) background noise from full-image extraction, and (iv) persistent polysemanticity in residual streams. One could extend the single-object concept extraction in CoX-LMM by extracting residual streams for sets of *(token, image)* pairs, but this remains inherently supervised and fails to overcome the aforementioned limitations.

While methods such as TCAV, CRAFT, and CLIP-Dissect pioneered concept-based interpretability, they target static encoders and cannot be applied to LVLMs. CoX-LMM incorporates many of these ideas into a dictionary-learning framework for autoregressive models, and thus serves as the most representative baseline. We therefore compare CGDL against CoX-LMM using its strongest dictionary learning variants (SNMF, SAE) to ensure fairness.

Unlike CoX-LMM and other concept extraction methods that rely on labeled tokens and often yield polysemantic vectors or assume single-object settings, we propose **Concept-Guided Dictionary Learning (CGDL)**. CGDL introduces contrastive residual extraction with spatially localized crops and candidate text, enforcing a clean separation between *concept* and *noise*. This produces faithful, monosemantic concept vectors and, to the best of our knowledge, is the first framework to scale weakly supervised concept discovery from single-object to multi-object settings.

## 3 PRELIMINARIES

### 3.1 DICTIONARY-LEARNING VIEW OF CONCEPT EXTRACTION

Recent work on concept extraction Ghorbani et al. (2019); Sun et al. (2023); Fel et al. (2023a); Parekh et al. (2024) shows that many methods can be framed as *dictionary learning*, where activations are approximated by a small set of interpretable bases.

Formally, given activations $S \in \mathbb{R}^{n \times d}$ with $n$ samples and $d$-dimensional features, we posit $K$ latent concepts and solve

$$\underset{U \in \mathbb{R}^{d \times K}, \, V \in \mathbb{R}^{n \times K}}{\arg\min} \|S - VU^\top\|_F^2,$$

where $U = [u_1, \ldots, u_K]$ are the **concept bases** (CAVs) and $V = [v_1^\top; \ldots; v_n^\top]$ are the **activations**, with row $v_i$ giving concept coordinates of sample $i$.

This factorization unifies prior approaches via constraints on $(U, V)$:

$$
\begin{cases}
v_i \in \{e_1, \ldots, e_K\}, & \text{K-Means (ACE) Ghorbani et al. (2019),} \\
U^\top U = I, & \text{PCA Graziani et al. (2023),} \\
S \geq 0, \, U \geq 0, \, V \geq 0, & \text{NMF (CRAFT Fel et al. (2023a;b), ,} \\
U \text{ free}, \, V \geq 0, & \text{Semi-NMF (SNMF) Trigeorgis et al. (2014); Parekh et al. (2024),} \\
V = \psi(S), \, \|v_i\|_0 \leq s, & \text{Sparse Autoencoder (SAE) Templeton (2024); Pach et al. (2025).}
\end{cases}
$$

Columns of $U$ are concept vectors (CAVs), rows of $V$ are per-sample activations. Special cases include PCA (orthogonal bases), NMF (nonnegative factors), SNMF (mixed-sign bases with nonnegative activations), K-Means (one-hot codes), and SAE (encoder-decoder with sparsity).

## 4 METHOD

We begin by formalizing LVLMs as black-box systems with accessible intermediate activations, taking multimodal inputs (text, image) and producing corresponding outputs (text/ set of tokens).

### 4.1 POSITIVE AND NEGATIVE CONCEPTS

We address the limitations of single-object dependence and polysemantic behavior in CoX-LMM by introducing *concept-example bags*—collections of image patches that serve as positive (concept-present) or negative (concept-absent) instances for each automatically discovered concept $c_k \in \{c_1, \ldots, c_K\}$.

Given an unlabeled image set $\mathcal{I} = \{I_k\}_{k=1}^N$, the LVLM $f$ predicts candidate concepts for each image using a structured prompt (See Appendix B for details.):

$$C(I_k) = f(I_k, \texttt{prompt}) \subseteq \mathcal{V},$$

and the global vocabulary is

$$\mathcal{C} = \bigcup_{k=1}^N C(I_k).$$

For each $c \in \mathcal{C}$, we collect supporting images

$$\Phi(c) = \{I_k \mid c \in C(I_k)\}.$$

A patch operator $\mathcal{P}(I_k, c) \in \{\mathrm{crop}(I_k, c), \mathrm{sam}(I_k, c)\}$ localizes the region associated with $c$ Kirillov et al. (2023). Because such patches may include background context, the resulting *concept-example bag* is a mixture of positives and negatives:

$$\mathcal{B}(c) = \{\,\mathcal{P}(I_k, c) : I_k \in \Phi(c)\,\} = \mathcal{B}^+(c) \cup \mathcal{B}^-(c).$$

Unlike prior approaches that rely on annotated single-object images, this formulation is *weakly supervised* and scales to multi-object datasets. For example, the concept "stripes" may emerge from zebras, tigers, or cats, without requiring manual concept labels.

### 4.2 CONCEPT-GUIDED DICTIONARY LEARNING

According to Fel et al. (2023a), most prior concept-expansion methods rely on dictionary learning for concept extraction; CoX-LMM is no exception. The key difference lies in the data passed to the dictionary-learning algorithm, which critically affects concept quality Grobrügge et al. (2025); Sun et al. (2023).

Relying on long, ambiguous token sequences from open-ended captioning typically restricts analysis to the residual stream of a single token, introducing overlapping concepts Templeton (2024) and preventing

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

**Reproducibility Statement** We release a reproducible pipeline requiring only a Hugging Face model card, access token, and dataset directory with `train`/`val` splits. CGDL and CoX-LMM can be run via `scripts/run_full_pipeline.sh` and `scripts/run_full_pipeline_dl.sh`, respectively. Code and configs are shared anonymously at `https://anonymous.4open.science/r/xl-vlms-30C1`, with installation and usage detailed in the `README`. All experiments used a single NVIDIA RTX 3090 (24GB) GPU with fixed random seeds.

**Ethics Statement.** This work poses no direct societal risks beyond those inherent to model interpretability. Our method may surface biased or harmful concepts present in LVLMs, which should be interpreted responsibly and not used to reinforce stereotypes.

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

## A  MONOSEMANTIC VS. POLYSEMANTIC REPRESENTATIONS

A central challenge in interpreting large vision–language models (LVLMs) lies in *superposition* and *feature entanglement* in high-dimensional residual streams (Elhage et al., 2022). Here, *features* can be understood as vector directions in activation space that encode candidate concepts. Ideally, such vectors should be *monosemantic*—each aligned with a single interpretable concept. In practice, however, LVLMs often learn *polysemantic* vectors, where a single direction is activated by multiple, semantically unrelated concepts.