# OpenReview forum: "Concept-Guided Dictionary Learning for Interpretable Concept Extraction and Attribution in Large Vision–Language Models"
_ICLR.cc/2026/Conference — Submitted to ICLR 2026_

### Official Review · Reviewer_Bj5C · 2025-10-31

**Soundness:** 2
**Presentation:** 2
**Contribution:** 2
**Rating:** 4
**Confidence:** 4

**Summary:**

This work proposes CGDL, a novel, weakly supervised method for discovering and attributing visual–textual concepts in autoregressive LVLMs. Unlike prior concept­-extraction techniques (e.g. TCAV, CRAFT, CLIP-Dissect), which assume static encoders or rely on labeled concepts in simplified settings, CGDL can scale to multi-object, open‐vocabulary scenarios. The key innovation is to prompt the LVLM with a binary “concept vs. no-concept” question (e.g. “Does this image contain concept cₖ?”), collect activation residuals on positive and negative patches, then solve a contrastive dictionary learning problem per concept. By decomposing each patch’s activation into a “concept” basis plus a residual basis, CGDL enforces monosemantic, sparse, and non-overlapping vectors that align visual regions with textual concepts. Experimentally on ImageNet-1k and MSCOCO, CGDL shows improvements over prior baselines in sparsity, stability, low overlap, and attribution faithfulness (e.g. +9 % in CLIPScore, +4 % BERTScore).

**Strengths:**

1. CGDL introduces a weakly supervised, scalable approach to concept extraction—a setting where most prior methods (e.g., TCAV, CRAFT, CLIP-Dissect, CoX-LMM) fail to generalize. By formulating concept discovery as a contrastive one-vs-all dictionary learning problem, CGDL avoids heavy supervision and scales effectively to thousands of visual–textual concepts.
2. The proposed two-basis factorization (concept vs. residual) and contrastive residual extraction lead to sparse, disentangled, and monosemantic concept vectors. This improves interpretability by ensuring that each discovered component corresponds to a single, semantically coherent concept rather than a polysemantic mixture.
3. Empirical results on ImageNet-1k and MSCOCO demonstrate stronger attribution faithfulness and higher quantitative interpretability metrics.
4. CGDL treats LVLMs as black-box systems, requiring only access to intermediate activations, and thus applies broadly to different architectures. Its per-concept two-basis decomposition reduces computational cost and avoids the atom collapse problem common in large-vocabulary dictionary learning, ensuring scalability without sacrificing quality.
5. Through contrastive concept prompts and SAM-based localization, CGDL effectively links visual patches with linguistic concepts, offering a coherent view of how LVLMs ground textual outputs in visual evidence. This multimodal grounding represents a key step toward transparent multimodal reasoning.

**Weaknesses:**

1. CGDL relies on SAM to construct positive and negative patch sets. While this provides weak supervision at scale, the quality of discovered concepts is sensitive to segmentation accuracy and crop relevance. In cluttered or fine-grained scenes, imperfect segmentation can lead to noisy or overlapping concept bags, weakening monosemanticity.
2. The method assumes that multimodal activations can be linearly decomposed into a “concept” and a “residual” subspace. This simplification may not capture the nonlinear dependencies or compositional reasoning inherent in LVLM representations, potentially limiting the interpretability of more abstract or relational concepts.
3. Although CGDL addresses autoregressive LVLMs, it focuses on single-layer residual activations and per-concept snapshots. It does not explicitly model how concepts evolve across generation steps or interact in temporal or contextual chains, which are key to understanding how LVLMs reason during long-form captioning or dialogue.
4. The experiments primarily report quantitative metrics (e.g., sparsity, overlap, CLIPScore, BERTScore) on ImageNet-1k and MSCOCO, which may not fully capture the qualitative interpretability or human alignment of discovered concepts. More extensive human studies or causal verification could strengthen claims about faithfulness and semantic clarity.
5. While the two-basis formulation improves efficiency, it may oversimplify concept structure when scaling to very large or nuanced concept vocabularies. Complex concepts that require multi-dimensional representations (e.g., “a person holding a red umbrella”) might not be well captured by a single basis vector per concept.

**Questions:**

1. How sensitive is CGDL to the assumption that activations can be linearly decomposed into concept and residual subspaces? Have you explored non-linear factorization methods (e.g., kernelized or autoencoder-based versions) to capture more complex multimodal interactions?
2. You analyze activations from the penultimate residual stream of the language model. Did you experiment with other layers or modalities (e.g., cross-attention maps) to assess whether concept alignment is layer-dependent?
3. The binary “concept vs. no-concept” prompting strategy is elegant, but could the binary framing bias activations toward token-level differences rather than broader semantic structures? How do you ensure it generalizes to abstract or relational concepts?
4. Since the framework depends on SAM for concept localization, how robust is CGDL to noisy or ambiguous segmentation masks? Would substituting SAM with another localization model (e.g., grounding DINO or CLIP-based proposals) materially change performance?

---

> ### Author Response · Authors · 2025-11-15
> **Clarifying CGDL’s sensitivity, layer choice, cross-attention, binary prompting, and faithfulness**
>
> **We thank the reviewer for the thoughtful question and for allowing us to clarify this aspect of our method.**
>
> Square brackets [ ] indicate references.
>
> ---
> ### **1. Sensitivity**
> We found the sensitivity of CGDL towards linear decomposition is low. We also explored the nonlinear factorization method, Sparse Auto Encoder (SAE).
> We assess CGDL’s sensitivity using the *stability* metric introduced in [a], where lower scores indicate lower sensitivity to noise in data. We sampled five subsets of concept bags and measured **stability **. CGDL consistently achieves lower (better) stability in both linear SNMF and nonlinear SAE; SNMF is more stable than SAE**, consistent with [b]. Results are reported in Table [2].
>
> Our linear decomposition choice is supported by prior works showing that concepts often align with linear directions in activation space, including T/NCAV [c,d] and NMF-based methods such as CRAFT [e,f].
>
> ---
> ### **2. Layer Dependence of Concept Alignment**
> We ran exploratory experiments on several mid-layer residual streams (layers 12–20) and found that, while concept vectors are still extractable, their grounding is noticeably unrelated: textual grounding becomes tokens and no longer matches visual grounding. This aligns with findings of systematic layer-wise study [f, h] and **prior evidence suggests that deeper layers in CNNs and transformers encode semantically rich and complex concepts [g,h].** These findings motivate our choice to operate on the penultimate residual stream.
> A hybrid variant that uses attention weights to reweight token representations is a promising direction for future work, but it is beyond the scope of this paper.
>
> ---
> ### **3. Token-Level Bias vs. Semantic Structure**
> CGDL uses binary prompts to separate image patches of a concept bag:
> (1) images **with** the concept, and (2) images **without** it.  For both, we query the *same* phrase (e.g., “hot dog”).
>
> The model outputs:
> - positive: `hot dog`
> - negative: `No hot dog`
>
> When computing residuals, we **drop `No`** and pool only over the shared tokens (*hot* + *dog*), so the text span is identical in both cases and only the **image** differs.
> | Case     | Image                  | Text output  | Residual span       | Contrast source              |
> |----------|---------------------|--------------|----------------------|------------------------------|
> | Positive | Contains hot dog | `hot dog`    | mean(*hot*, *dog*)   | Visual features support cₖ   |
> | Negative | No hot dog present   | `No hot dog` | mean(*hot*, *dog*)   | Visual features reject cₖ    |
>
> Thus, SNMF sees two clouds of residuals over the *same tokens* and must separate them based on **image-conditioned activations**, not token-level differences, yielding the concept vector that encodes visual presence vs. absence of \(c_k\).
>
> We apply mean pooling [i] to aggregate residuals over all tokens in the concept span (Sec 4.2 Eq. 3 in manuscript) and then learn concept vector directions on this pooled representation. This pooling helps capture abstract concepts (e.g., *hot dog*) at the span level rather than relying on any single token *hot* or *dog* individually.
>
> ---
> ### **4. Robustness to SAM and alternative localizers.**
> CGDL does **not** depend on precise SAM masks: SAM only proposes possible candidate regions for concept vector extraction, while the LVLM’s identifies the content of the crop.
>
> Replacing SAM with **Grounding DINO** on 10 MSCOCO classes (same setup as Table 3 in the manuscript) yields very similar CLIPScore/BERTScore:
>
> | Metric   | Random        | Text-only    | Image-only      | Combined        |
> |----------|--------------:|-------------:|----------------:|----------------:|
> | CS top-1 | 0.50 ± 0.04   | --          | 0.60 ± 0.07     | 0.64 ± 0.06     |
> | BS top-1 | 0.71 ± 0.01   | 0.91 ± 0.08  | 0.90 ± 0.06     | 0.92 ± 0.08     |
>
> We avoid CLIP-based proposals to not bias CLIPScore, but they are a drop-in replacement. **We compare the result of SAM vs random and give some example output in Figures 5 and 6 and an ablation table in the appendix (Table 10).**
>
> ---
> ### **5 Faithfulness test (W4)**
> We use a standard unsupervised XAI faithfulness test (Sec. 5.3, Fig. 1) based on well-established concept deletion/addition curves [a,e]. High-activation concepts cause sharp probability changes when deleted or added, while lower-ranked ones have smaller effects; CGDL shows this pattern much more clearly than CoX-LMM, indicating better alignment with the model’s decisions. Human studies are valuable future work, but beyond this paper’s scope.
>
> ### **References**
>
> [a] Fel et al., Holistic, NeurIPS 2023.
>
> [b] Fel et al., Archetypal SAE, ICML 2025.
>
> [c] Kim et al., TCAV, ICML 2018.
>
> [d] Zhang et al., NCAV, AAAI-21.
>
> [e] Fel et al., CRAFT, CVPR 2023.
>
> [f] Parekh et al., CoX-LMM, NeurIPS 2024.
>
> [g] Alain & Bengio, Linear Probes, 2016.
>
> [h] Jin et al., Exploring Concept Depth, COLING 2025.
>
> [i] Soler et al., Word Splitting & Semantics, TACL 2024.

---

> > ### Comment · Reviewer_Bj5C · 2025-11-24
> >
> > Thank you to the authors for their response. I still find my original overall score appropriate.

---

> ### Author Response · Authors · 2025-11-28
>
> **We would like to provide further clarification, as we have conducted additional experiments.**
>
> **Additional answer to Question 2:**
>
> In the meantime *we ran a layer-wise ablation study (Figure 12, Section F) for Gemma-3n*. The findings are consistent with previous research (CoX-LMM), which found that CLIPScore tends to be higher in deeper layers and lower in early (shallower) layers.
>
> Given that CLIPScore concentrates on high-level image–text concept similarity, this makes sense. While deeper layers encode more global, semantic information and thus attain higher CLIPScores, early layers primarily capture local, low-level features, making their alignment with text weaker.
>
> We do not observe a relationship between model depth and BERTScore. There is no consistent increasing pattern in the BERTScores given by different layers. This layer independence trend is expected because text does not naturally split into "low-level" and "high-level" features like images do, and BERTScore measures text-text similarity. Overall, these results imply that while some layers yield concepts that are less semantically meaningful, other layers are particularly helpful for concept extraction (they produce concepts that align well with ground-truth descriptions). However, there is currently no reliable metric for comparing text to low-level visual features, which suggests an intriguing avenue for further research on low-level feature-to-text similarity.
>
> **Additional answer to Question 3:**
>
> *Our approach purposefully aggregates hidden embeddings across all tokens that comprise a concept, in contrast to baselines that depend on the hidden state of a single token.*
>
> A concept is written as a complete phrase in our binary prompts, such as "person holding a red umbrella" versus "no person holding a red umbrella." Next, we obtain a single concept embedding by applying mean pooling (Eq. (3), Section 4.2) to the hidden representations of each token in this phrase. This design enables us to represent multi-token concepts like "brown bear" or "hot dog," avoiding the token-level restriction of previous approaches.
>
> We concentrate on monosemantic concepts, where each concept corresponds to a single underlying idea rather than a combination of multiple concepts, so using a long sentence as a concept is outside the scope of this paper.
>
> The majority of concepts are naturally expressed as multi-token phrases if one examines the textual groundings above the concept images in the qualitative examples (Figure 5 in the appendix). Figures 9 and 10 in the Appendix provide more qualitative examples of abstract, multi-token, and even related concepts from different objects.

---

> > ### Author Response · Authors · 2025-12-01
> > **Summary comment**
> >
> > We thank the reviewer. We would be happy to engage in any follow-up discussion or clarification if circumstances allowed it.
> > We have answered all the questions raised and clarified the reviewer’s concerns.
> >
> > - **Sensitivity:** Table 2 in the Results section shows that CGDL is less sensitive to the choice of decomposition. Our linear SNMF setup is more stable than the nonlinear sparse autoencoder–based variant.
> >
> > - **Layer choice:** We added layer-wise ablations on Gemma-3n (Figure 12, Section F), which support using the penultimate residual stream for concept extraction.
> >
> > - **Token vs. semantic structure:** Section 4.2 (Eq. (3)) explains our binary prompting and span-level mean pooling, which enable multi-token concepts. This is one of the strengths of our method compared to related approaches.
> >
> > - **Robustness & faithfulness:** Experiments with different localizers (Figures 5–7, Table 10 in the appendix) and our deletion/addition-based faithfulness test (Section 5.3, Figure 1) show that CGDL is better aligned with model decisions than recent XAI methods for LVLMs.

---

### Official Review · Reviewer_AqDc · 2025-10-31

**Soundness:** 3
**Presentation:** 2
**Contribution:** 2
**Rating:** 6
**Confidence:** 2

**Summary:**

The article addresses the pain point of verifying whether autoregressive Large Vision-Language Model (LVLM) predictions are grounded in visual concepts or result from bias or hallucination. The proposed method, Concept-Guided Dictionary Learning (CGDL), is designed for interpretable concept extraction and demonstrates consistent performance gains over the prior art (COX-LMM). A core experimental result highlights CGDL's effective scalability from 10 to 1k concepts, achieving a strong Combined Score of $0.64 \pm 0.06$ on the ImageNet 1k concept set when using combined text and image conditioning

**Strengths:**

The research exhibits several notable strengths. First, the work tackles the critical and challenging problem of concept grounding and attribution within autoregressive Large Vision-Language Models (LVLMs), successfully extending interpretability methods beyond simpler encoder-only architectures. Second, the proposed Concept-Guided Dictionary Learning (CGDL) framework demonstrates robust scalability, showing consistent performance gains when moving from small-scale (10 concepts) to large-scale (1k concepts) benchmarks, which is essential for real-world concept dictionaries. Third, the empirical results are clearly quantified and rigorously benchmarked, with CGDL achieving a high Combined Score (CS) of $0.64 \pm 0.06$ on the ImageNet 1k concept set, providing strong evidence of its attribution effectiveness.

**Weaknesses:**

Despite its strengths, the work presents a few limitations that warrant further discussion. The generalizability of the CGDL method may be limited as the current results primarily emphasize performance relative to COX-LMM; hence, verification on a broader spectrum of distinct LVLM architectures is needed. Additionally, the rationality of core design choices in the dictionary learning process, such as the selected dictionary size or the specific sparsity constraints, requires a more detailed sensitivity analysis to demonstrate that the framework is robust across hyperparameter variations. Furthermore, the unexplored direction of concept types could be addressed by extending the evaluation beyond the current visual concepts (ImageNet 1k) to include more abstract or relational concepts.

**Questions:**

Could the authors provide a more extensive ablation study detailing the influence and stability of the core hyperparameters governing the dictionary learning process within CGDL, offering empirical evidence to justify the final selection of key parameters? I also recommend including a discussion of the method’s performance and stability when applied to different foundational LVLM architectures or when tasked with attributing more abstract, non-object concepts critical for complex reasoning to confirm broader generalizability.

---

> ### Author Response · Authors · 2025-11-17
> **Data size, dictionary size, and LVLM choice**
>
> We thank the reviewer for giving valuable comments and giving the opportunity to add more experimental results.
>
> - We conducted two ablation studies on Gemma-3n using five sampled ImageNet classes: (i) dictionary size K (number of atoms), and (ii) SNMF sparsity weight alpha with fixed K. The results show that CLIPScore is notably more sensitive to K than to alpha: small dictionaries (K=2,10) achieve the best CLIP@1, while larger K generally reduces CLIP@1 even though BERT@1 stays nearly constant. In contrast, varying alpha causes only minor CLIP@1 changes within the standard deviation, so we keep a moderate sparsity setting (alpha=20) in the main experiments.
>
> **Dictionary size K ablation (Top-1)**
>
> | K | BERT@1 ↑ | CLIP@1 ↑ |
> |---:|:---:|:---:|
> | 2   | 0.881 ± 0.013 | 0.616 ± 0.044 |
> | 10  | 0.881 ± 0.013 | 0.603 ± 0.047 |
> | 30  | 0.884 ± 0.040 | 0.475 ± 0.033 |
> | 50  | 0.885 ± 0.040 | 0.509 ± 0.028 |
> | 100 | 0.887 ± 0.041 | 0.488 ± 0.035 |
>
> **SNMF alpha ablation (Top-1)**
>
> | alpha | BERT@1 ↑ | CLIP@1 ↑ |
> |---:|:---:|:---:|
> | 0   | 0.881 ± 0.013 | 0.610 ± 0.039 |
> | 20  | 0.881 ± 0.013 | 0.616 ± 0.044 |
> | 100 | 0.881 ± 0.013 | 0.615 ± 0.043 |
> | 150 | 0.881 ± 0.013 | 0.617 ± 0.043 |
> | 200 | 0.881 ± 0.013 | 0.621 ± 0.039 |
>
> The full result is written in Section E ABLATION STUDY of the Appendix.
> - In the appendix, two additional result tables were presented using datasets with different numbers and sizes of images: DTD (Describable Texture Dataset, which is more low-level texture than object/class) and CIFAR-100. \
> DTD contains more abstract texture patterns than standard object datasets, and our method performs attribution on DTD comparably well. **Figure 10 in Section E.5 in the Appendix shows the visual explanation of model prediction and explanation using abstract concept, such as color, pattern, etc.** On the other hand, the baseline paper explains predictions only using objects from the same class, and it does not demonstrate how these explanations relate to more abstract visual concepts.
>
> - We will highlight in the discussion how the dictionary size affects our method’s performance. Table 2 already shows that when we increase the dictionary size from 10 to 1000, there is a slight drop in dictionary evaluation metrics. We also observe that there is very little change in CLIPScore and BERTScore as the dictionary size increases.
>
> - Different LVLM architectures. In addition to Gemma-3n-4B, we ran our experiments on Qwen2-7B-VL and Qwen2.5-7B-VL (In Appendix). These models differ in parameter count, training strategy, and residual dimensionality. We observe that the 4B-parameter **Gemma-3n-4B performs almost the same as Qwen2.5-7B-VL on the attribution tasks**. Qwen2-7B-VL, an earlier model, performs slightly worse than Qwen2.5-7B-VL and Gemma-3n-4B. We hypothesize that this is because newer models use more sophisticated training approaches; even though Gemma-3n-4B has only 4B parameters, training with knowledge distillation likely helps it reach concept attribution performance comparable to the 7B Qwen models.
>
> | Model          | Params | Hidden Size | Layers | Release Date | Distilled? |
> |----------------|--------|-------------|--------|--------------|-----------|
> | Gemma-3n-4B    | 4B    | 2,048       | 35     | Jun 2025     | Yes       |
> | Qwen2.5-7B-VL  | 7B    | 3,584       | 28     | Jan 2025     | No        |
> | Qwen2-7B-VL    | 7B    | 3,584       | 28     | Aug 2024     | No        |

---

> > ### Comment · Reviewer_AqDc · 2025-11-25
> >
> > Thank you for the efforts dedicated to the rebuttal. I shall retain my original evaluation score.
> >
> > Furthermore, I think it will be better if the authors could present the key conclusions derived from the appendix and explicitly cross-reference the corresponding appendix sections in the main text. Currently, only 'Appendix B' and 'Appendix E' are cited in the main body, while references to other appendix sections / key conclusions appear to be absent.

---

> > > ### Author Response · Authors · 2025-11-28
> > >
> > > Thank you again. We have now cross-referenced the findings in the appendix with the main text.

---

> > > > ### Author Response · Authors · 2025-12-01
> > > > **Summary comment**
> > > >
> > > > We addressed all questions raised by the reviewers in our rebuttal and revisions:
> > > >
> > > > - We added ablation studies on dictionary size ($K$) and sparsity ($\alpha$) (see **Appendix F: Ablation Study**), showing that CGDL is robust and justifying our default hyperparameter choices.
> > > > - We extended experiments to a more abstract concept dataset (e.g., Describable Textures Dataset (DTD)) and to additional LVLM architectures (Gemma-3n-4B, Qwen2-7B-VL, Qwen2.5-7B-VL), with detailed results provided in **Appendix Sections E.1 and E.2**.
> > > > - We added qualitative explanations using related and abstract concepts, with examples shown in **Figure 9** and **Figure 10** in the appendix.

---

### Official Review · Reviewer_wx7H · 2025-10-31

**Soundness:** 2
**Presentation:** 1
**Contribution:** 3
**Rating:** 6
**Confidence:** 4

**Summary:**

This paper addresses the challenge of discovering interpretable and faithful concepts in autoregressive LMMs, where predictions may stem from hallucinations or spurious correlations rather than grounded visual evidence. To this end, the authors propose CGDL, a weakly supervised framework that automatically constructs positive and negative concept examples using segmentation-based localization and performs contrastive dictionary learning to disentangle concept-aligned activations from residual noise. By introducing a two-basis decomposition and monosemantic regularization, CGDL yields sparse, disentangled, and multimodally aligned concept representations. Extensive experiments demonstrate that CGDL achieves superior sparsity, stability, and faithfulness compared to existing interpretability methods for LVLMs.

**Strengths:**

* This paper takes an important step by directly modeling concepts as generative factors within LMMs. I am happy that compared with existing post-hoc explanation methods the proposed approach provides stronger guarantees for model interpretability.
* The authors discuss the related literature in considerable detail.
* The paper is easy to follow.
* The authors provide code to facilitate reproducibility checks.

**Weaknesses:**

1. Please discuss the training cost of the proposed method and inference efficiency (e.g., FLOPS and memory usage compared to the LMM backbone), and compare the computation efficiency with existing approaches, if possible.
2. It is recommended that the authors include more visualizations of the learned concepts and analyze cases of both correct and incorrect model reasoning to better illustrate the advantages and limitations of the proposed method compared with prior works.
3. This paper is poorly written, with incorrect citation formatting and numerous punctuation issues, including inconsistent and incorrect use of dashes and hyphens.

**Questions:**

My questions are in Weaknesses Section.

---

> ### Author Response · Authors · 2025-11-17
>
> We thank the reviewer for highlighting some important questions and suggestions. We tried our best to answer the reviewer's questions.
>
> We appreciate the proposal of the discussion about the cost in terms of FLOPS, and we also ran an experiment for MSCOCO 10 objects in Gemma-3n-4B to check the computation time.
>
> ### **1. Computation cost:**
>
>  Our method needs 421 seconds for 10 concepts, while CoX-LMM needs 4375 seconds for 10 concept extractions, given the images are mapped with the concepts (concept bag) in GPU: NVIDIA RTX 3090 (24GB) CPU: AMD Ryzen 9 5950X 16-Core Processor
> **CGDL FLOPs $\approx 4.28 × 10^{16} $** \
> **CoX-LMM (baseline) FLOPs $ \approx 7.20 × 10^{16} $**
>
> *At the bottom of the comment section, we provide a detailed analysis of the FLOPs required for concept extraction and inference. Please feel free to ask any questions.*
>
> ---
>
> ### **2. Visualization for qualitative analysis**
>
> **We added more visualizations of learned concepts and LVLM post-hoc explanations in Appendix E.3, E.4, E.5, and E.6, including an example from prior work (Figure 6 in the appendix) and several failure cases.**
> Our method can explain the LVLM’s prediction token by token. For example, given a 2×2 image grid, we prompt the LVLM to detect objects in the grid, and CGDL then explains each generated token using the corresponding concept activations.
>
> **We also highlight a failure case in Figure 10 of the appendix**. Occasionally, the textual grounding of a concept shifts to a less-related token, while the visual grounding remains accurate and informative.
>
> ---
>
> ### **3. Writing issues**
>
> We thank the reviewer for pointing this out. We promise we will revise the paper to improve writing quality, correct citation formatting, and fix punctuation issues in the camera-ready version.
>
> ---
> Training and inference cost: CGDL vs. existing method:
>
> | Category        | CGDL                               | CoX-LMM                            | Comment |
> |-----------------|-------------------------------------|-------------------------------------|---------|
> | **Training GPU FLOPs** | $$4.28 \times 10^{16}$$          | $$7.20 \times 10^{16}$$              | CGDL$ \approx$ 40% cheaper |
> | **Training CPU Ops**   | $$3.28 \times 10^{10}$$          | $$1.43 \times 10^{11}$$              | CoX-LMM  $\approx$ 4× higher |
> | **Inference FLOPs / image** | $$1.6 \times 10^{11}$$ (same) | $$1.6 \times 10^{11}$$ (same)        | Cosine cost negligible |
> ---
>
> ### **Detailed Analysis of FLOPs**
>
> The computational cost is divided in terms of training and inference.
>
> * Training cost:  Generating residual activations with the LVLM backbone + SNMF training cost (CPU): learning the concept dictionaries from these activations.
>
> * Inference cost: Standard VLM backbone inference cost + cosine distance computation between residual activations and concept vectors to map the most similar concepts.
>
> **FLOPs Estimate for Gemma-3n-4B on MSCOCO-10 Objects**
>
> Number of model parameters: $M = 4 × 10^9$
>
> FLOPs per LVLM forward pass: FLOPs/sample $ = 8 × 10^9$ (~2 times model size)
>
> T = total token length (image + text + outputs).
>
> ***CGDL***
>
> ### *(1) Concept bag creation (Sec 4.1)*
>    Number of images: N_img = $3000 $ ($300$ images/ category)\
>    Token length per sample: $T = 196 + 40 + 200 = 436$\
>     FLOPs_imgs = FLOPs/sample × T × N_image ≈ $1.05 × 10^{16}$
>
> ### *(2) Segmentation cost (SAM)  (Sec 4.1)*
>
>    Total images: $3000$
>
>    Crops: $16000$ crops
>
> SAM FLOPs: FLOPs_SAM = $1.1 × 10^{12} × 3000 = 3.3 × 10^{15}$
>
> ### *(3). Extracting residual streams for concept bags (Sec 4.2)*
>
> Number of residual samples: N_res = $16000$ ($1600$ crops/ concept)
>
> Token length during binary questioning :
> T_res  $ \approx196 + 21 + 10 = 227$
>
> Residual FLOPs:
> FLOPs_res $\approx 8 × 10^9 × 227 × 16000 \approx 2.91 × 10^{16}$
>
>
> ### *(4). SNMF  (CPU) computation (Sec 4.2)*
>
> Parameters: $N = 1600$,  $D = 2048$,  $K = 2$,  $iters = 5000$ , $Concepts = 10$
>
> CPU operation: Ops_CGDL,SNMF $\approx 1,600 × 2,048 × 2 × 5,000 × 10 = 3.28 × 10^{11}$
> 3. Total LVLM GPU computation (Sec 4.2)
>
> **FLOPs_CGDL,total $\approx 1.05 × 10^{16} + 3.3 × 10^{15} + 2.91 × 10^{16}  = 4.28 × 10^{16}$**
>
> ***Existing method, CoX-LMM***
>
> ### *(1). Caption string search (CPU)*
>
> Finding images of $10$ objects in the whole dataset caption using regex (minimum $120,000$ captions (MSCOCO) for checking the whole dataset, approximately $400$ chars per caption including space)
>
> Ops_CoX,search $\approx 120,000 \times 400 = 4.2 × 10^8$
>
> ### *(2). Extracting residual streams*
>
> Number of images: $35,000$ ($\approx 3,500$ images per object, $10$ objects)
>
> Token length:
> T_res $\approx 196 + 50 + 11 = 257$
>
> Residual FLOPs:
> FLOPs_CoX, res $ \approx 8 × 10^9 × 257 × 35,000 \approx 7.196 × 10^{16} $
>
> **Total LVLM FLOPs:**
> FLOPs_CoX, total $\approx 7.20 × 10^{16}$
>
> ### *(3). Dictionary learning  (CPU)*
>
> Parameters: N = 35,000, D = 2,048, K = 10, iters = 200
>
> Ops: Ops_CoX, SNMF $\approx  35,000 × 2,048 × 10 × 200 = 1.43 × 10^{11}$

---

> > ### Author Response · Authors · 2025-12-01
> > **Summary comment**
> >
> > In our response to the reviewers’ comments, we addressed all questions and made the following updates and clarifications:
> >
> > 1. **Training cost and inference efficiency**
> >    We compared the training cost and inference efficiency of our proposed method with the existing approach. Both theoretically and experimentally, we showed that our method is less expensive. Further details are provided in **Appendix Section G**.
> >
> > 2. **Post-hoc explanations and qualitative analysis**
> >    We added extensive post-hoc explanations for qualitative analysis in **Appendix Sections E.3, E.4, and E.5**. In **Figure 5** (our approach) and **Figure 6** (existing approach) in the appendix, we compare the outputs and show that our method provides more meaningful explanations of a Large Vision-Language Model.
> >
> > 3. **Citation and punctuation corrections**
> >    We corrected citation formatting and punctuation issues in the revised version.

---

### Official Review · Reviewer_PnqH · 2025-10-31

**Soundness:** 2
**Presentation:** 3
**Contribution:** 2
**Rating:** 2
**Confidence:** 3

**Summary:**

This paper extends the CoX-LMM (NeurIPS 2024) method for concept-based explainability of vision-language models, such as LLaVA, Gemma, and Qwen-VL. It improves basic metrics for concept extraction by leveraging the Segment Anything Model (SAM) and Concept Activation Vectors (CAV). The experimental evaluation features qualitative examples that demonstrate the method's ability to discover fine-grained, monosemantic concepts, as well as its utility in attributing concepts to the model's responses.

**Strengths:**

The key strengths of the paper are:
1. Convincing motivation, i.e. we need good explanation methods for large vision-language models.
2. Comprehensive experiments, spanning multiple models, metrics, and applied use-cases.
3. Good writing and presentation (Figures, Tables).

**Weaknesses:**

The main result of this work is improving a few metric values by a bit with a complex methodology stitched from known approaches; not gaining new insights or discoveries:
1. The proposed method is overcomplex; it includes four steps (Sec. 4.1-4.4), which have already been proposed (countless references to related work and reusing known ideas), making it challenging to evaluate the paper's contribution. For example, how is labeling concepts with an LVLM novel? See e.g. [a,b]. Or using the Segment Anything model in the process; see e.g. [c]. Sections 4.3 and 4.4 already begin with "Following Parekh et al. (2024)/Kim et al. (2018)" and so on.
2. This paper heavily builds on the previous work "CoX-LMM (Parekh et al., 2024)", mentioning it 36 times, and compares only to it in experiments. The experimental setting (dataset, metrics, etc.) is specific to these two methods, making the contribution less significant.

[a] Label-free concept bottleneck models. ICLR 2023

[b] Language in a bottle: Language model guided concept bottlenecks for interpretable image classification. CVPR 2023

[c] DCBM: Data-efficient visual concept bottleneck models. ICML 2025

**Questions:**

1. The word "scalable" is used 6 times in the paper; how is it measured? Similarly, "flexible" and "efficient" appear without any supporting evidence.
2. Why are the three highlighted metrics: "4% higher sparsity, 11% greater stability, 17% lower overlap" important for interpretability?
3. Why are the experiments conducted using the Gemma and Qwen models, when CoX-LMM was evaluated with the DePALM and LLaVA models? This mismatch is confusing.

- There is just too much bold and italicized text in the paper, making it hard to read.
- In Table 1: "Unlimited-object" listed as a "Key Limitation" sounds odd.

---

> ### Author Response · Authors · 2025-11-14
> **Clarifying the Core Goal and Answering the Questions**
>
> **We thank the reviewer for the detailed and constructive feedback. We address your question below after clarifying the misunderstanding.**
>
> ### **1. Goal of the paper**
> We regret that our contribution was not clearly visible. We will clarify it in the revision.
> Our method fundamentally differs from [a, b, c].
> We directly explain the LVLM’s prediction using concept vectors and their example images. (We added illustrative figures in Appendix E.3 for clarity.) In contrast, [a, b, c] use the LVLM/LLM only to generate concepts, which are then used to train a separate one-layer interpretable model for explaining another image classification model.
>
> *We extract direction vectors in the LVLM’s hidden space that the model uses during autoregressive token generation, and we leverage the vectors for explaining the LLVM prediction in a post-hoc manner.*
>
> The main novelty of our method lies in Sec. 4.1–4.2. We first use the LVLM to discover which concepts are present in the image dataset and build a concept -> image mapping. Since each image may contain multiple concepts (including background), we then apply SAM and random cropping to separate relevant from irrelevant regions, producing positive and negative concept examples.
>
> Next, we use a binary prompting procedure to extract the hidden activations for both positive and negative samples. We then apply a binary linear or nonlinear decomposition to separate these activations and obtain two representative vectors: one positive vector (the concept vector) and one negative vector. The positive vector becomes the dictionary key, and the corresponding positive images form the values. By repeating this process for all concepts, we obtain a full concept dictionary for all concepts the model implicitly knows in the dataset. Negative examples are discarded.
>
> At inference, we explain model predictions by computing the cosine similarity between the model’s activation and the dictionary keys, retrieving the concepts that best match the prediction.
>
> #### **The role of SAM**
>
> SAM is *not* part of the core algorithm; it is used only once as a lightweight pre-filter to separate background patches (L181–185).
> Random cropping is also an alternative but noisier because crop locations are random.
>
> ---
>
> ### **2. Why comparison is only to CoX-LMM**
>
> To our knowledge, CoX-LMM is the only prior work that performs post-hoc concept extraction for LVLMs; all other concept-based methods target CNNs with non-autoregressive activations. Our method is semi-supervised, whereas CoX-LMM is fully supervised. Our approach also produces more localized, concept-specific regions with less background noise, while the SOTA often captures large background areas. Finally, we evaluate on four datasets (listed in the appendix), whereas CoX-LMM uses only 10 ImageNet classes.
>
> ---
> **Answers to the Reviewer’s Specific Questions**
>
> ---
> ## **1. Clarifying “scalable,” “flexible,” and “efficient”**
>
> We will drop “flexible’’ and “efficient’’ to avoid overstating claims.\
>
> **Scalable:**
> CoX-LMM must be run per object class, producing many redundant vectors (e.g., repeated “fur’’ concepts). Our method performs cross-object extraction, yielding one shared concept vector.
>
> **Efficient (Kept only for explanation)**
> We use only two vectors per concept, while CoX-LMM needs multiple cluster vectors. Our binary questions also require ~10 output tokens per image, whereas CoX-LMM depends on long captions to obtain embeddings.
>
> ---
> ### **2. Importance of sparsity, stability, and overlap**
>
> These metrics are standard indicators of interpretable concept vectors [e,f]:
>
> **Higher sparsity**: Activates only under specific hidden states, producing a sharp, focused relation between the prediction and the concept, rather than a diffuse one.
> **Greater stability**:  Concept directions remain stable under small changes in the input. For example, the appearance of a nose in the image will not affect the explanation.
> **Lower overlap**: Each concept vector is separated from the others, with each one corresponding to a distinct input factor.
>
> **We will clarify this explicitly in the revision.**
>
> ---
>
> ### **3. Why Gemma and Qwen instead of DePALM and LLaVA**
>
> Our work targets fully autoregressive LVLMs, where visual and text tokens share a unified embedding space across all layers, making it important to understand how visual information flows through generation. In contrast, DePALM and LLaVA use bridge networks that convert visual features into LLM tokens, limiting access to how visual evidence actually influences the autoregressive process.
>
> ---
>
> ### **Formatting**
>
> We will reduce bold/italic usage and correct the location of the “Unlimited-object’’ entry (it is an advantage, not a limitation).
> ### **Reference List (for reviewer convenience)**
> [a] Label-Free CBM, ICLR 2023
> [b] Language-in-a-Bottle, CVPR 2023
> [c] DCBM, ICML 2025
> [d] CoX-LMM, NeurIPS 2024
> [e] CRAFT, NeurIPS 2020
> [f] Holistic Concept Extraction, ICML 2022

---

> > ### Comment · Reviewer_PnqH · 2025-11-22
> > **Thank you for clarifications**
> >
> > I thank the Authors for answering my questions and improving the manuscript, although the two weaknesses remain unresolved.
> >
> > The core of the proposed method is to prompt an LVLM to label concepts in images (Section 4.1 & Appendix B "Concept generation prompt"), and then prompt the LVLM again to extract concept embeddings from images (Section 4.2 & Appendix B "Concept-Guidance prompt"). This heavy reliance on an arbitrary prompt, as well as the particular LVLM's accuracy in executing it correctly, **is another limitation**. It is also not clearly presented in the Abstract and Introduction.
> >
> > ----
> >
> > **W1 (TL;DR): The proposed method is overly complex, and the novelty is unclear.**
> >
> > > Our method fundamentally differs from [a, b, c].
> >
> > Yes, these were only examples for the specific parts of the methodology.
> >
> > > We directly explain the LVLM's prediction using concept vectors and their example images.
> >
> > Concept activation vectors have been extensively used to explain both LLMs and vision models. They can be adapted to LVLMs without introducing overly complex methodology, or at least as a baseline to compare with it. It becomes common to interpret and steer LVLMs with sparse autoencoders extracting concepts [g, h].
> >
> > > We added illustrative figures in Appendix E.3 for clarity.
> >
> > Thank you, these figures represent a valuable way of evaluating the proposed methodology (btw. they resemble the pointing game evaluation from [i]).
> >
> > > In contrast, [a, b, c] use the LVLM/LLM only to generate concepts, which are then used to train a separate one-layer interpretable model for explaining another image classification model.
> >
> > > The main novelty of our method lies in Sec. 4.1–4.2. We first use the LVLM to discover which concepts are present in the image dataset and build a concept -> image mapping.
> >
> > These two statements seem to contradict each other. It is not necessarily novel to just prompt an LVLM to label concepts in images [a, b, h, j].
> >
> > > **The role of SAM.** SAM is not part of the core algorithm; it is used only once as a lightweight pre-filter to separate background patches (L181–185). Random cropping is also an alternative but noisier because crop locations are random.
> >
> > Untrue. Using SAM is stated as a core contribution of this paper in L79.
> >
> > **W2 (TL;DR): This paper heavily builds on the previous work, CoX-LMM [d], making the contribution less significant.**
> >
> > > To our knowledge, CoX-LMM is the only prior work that performs post-hoc concept extraction for LVLMs; all other concept-based methods target CNNs with non-autoregressive activations. [...]
> >
> > Can you clarify how this statement relates to the fact that sparse autoencoders have been used to extract concepts in LVLMs, e.g. [g, h]? Not to mention extracting concepts from ViT-based CLIP models..
> >
> > ----
> >
> > **Q3: Why are the experiments conducted using the Gemma and Qwen models, when CoX-LMM was evaluated with the DePALM and LLaVA models? This mismatch is confusing.**
> >
> > > Our work targets fully autoregressive LVLMs, where visual and text tokens share a unified embedding space across all layers, making it important to understand how visual information flows through generation. In contrast, DePALM and LLaVA use bridge networks that convert visual features into LLM tokens, limiting access to how visual evidence actually influences the autoregressive process.
> >
> > This mismatch remains confusing and should be clarified in the Introduction and Preliminaries/Methodology, particularly given the discussion in **W2**.
> >
> > **Reference List (copied & extended)**
> >
> > [a] Label-Free CBM, ICLR 2023
> >
> > [b] Language-in-a-Bottle, CVPR 2023
> >
> > [c] DCBM, ICML 2025
> >
> > [d] CoX-LMM, NeurIPS 2024
> >
> > [e] CRAFT, NeurIPS 2020
> >
> > [f] Holistic Concept Extraction, ICML 2022
> >
> > [g] Sparse Autoencoders Learn Monosemantic Features in Vision-Language Models, arXiv:2504.02821, NeurIPS 2025
> >
> > [h] Large Multi-modal Models Can Interpret Features in Large Multi-modal Models, arXiv:2411.14982, ICCV 2025
> >
> > [i] Explaining Similarity in Vision-Language Encoders with Weighted Banzhaf Interactions, arXiv:2508.05430, NeurIPS 2025
> >
> > [j] Pre-trained Vision-Language Models Learn Discoverable Visual Concepts, arXiv:2404.12652, TMLR 2025
> >
> > ----
> >
> > **Side note:** It would be helpful if you distinguish changes in the manuscript's text with blue color, etc.

---

> ### Author Response · Authors · 2025-11-24
> **Further clarification**
>
> **Prompt**
>
> Our prompts are deliberately simple and are handled reliably by common LVLMs, including models smaller than 10B; for example, Gemma-3n (4B parameters) follows them consistently in our experiments. We therefore did not initially expect prompt misunderstanding to be an issue. Nevertheless, we now note prompt dependence and potential failures on some LVLMs as a limitation in the Abstract and Conclusion. For the binary prompt, we provide an ablation study (Table 4) showing that variations in wording do not harm performance.
>
> ---
>
> **CAV as Baseline**
>
> An LVLM generates output tokens autoregressively, and for each predicted token, it produces a distinct hidden representation at every layer. For example, given an image of a black dog, the model might generate the caption "In this picture there is a black dog."; each token in this sequence is associated with a different internal activation pattern. CoX-LMM adapts CAVs to LVLMs by extracting hidden activations corresponding to a "specific" target token (e.g., "dog") and using that token as a semantic handle for the visual concept. This design, however, has two important limitations: (i) the object of interest must be explicitly named in the generated output, and (ii) for multi-token concepts (e.g., ``hot dog'') the explainer has to pre-select which token or subworld to use (the original authors suggest using the last subtoken), which is a fragile and ad-hoc choice. So, we used CoX-LMM as the baseline for comparison, as it already performed post-hoc concept-based explosibility for the Vision Language Model using the CAV idea.
>
>
> ----
> **Difference between CBM and CGDL**.
> We agree that our first step in 4.1, prompting an LVLM to surface what concepts exist in the dataset, is related to prior work that discovers concepts from images using LVLMs in papers [a, b, h, j]. We added it in the related work and clearly mentioned why our method is different from this related work, as we answered in the previous session.
>
> ----
>
>
> **Sam vs Random Cropping only**
> We added visual examples for SAM vs Random in the appendix for further identifying the importance of SAM. Adding SAM indeed helpful for visualization. We are running an ablation study to see the actual difference between SAM and random crops. We will update when the result is ready. For clarification, we added a statement in Section 4.1, we are using SAM for localizing crops for explaining LVLM, while [c] uses SAM to develop the CBM model. The intent is quite different.
>
>
>
> **SAE for feature extraction and interpretability**
>
> [h] is very recent, and we mistakenly omitted it; [g] is already discussed in related work. We find SAEs promising for mechanistic feature discovery and model steering, so we see them as complementary. However, for strict post-hoc explanation, we prefer not to modify the target model. Methods like [g, h] insert an SAE module into the forward pass, effectively adding a new nonlinear layer that can change the activation space and alter the model’s original accuracy—this limits their suitability for post-hoc interpretation. We have added both methods and clarified how they differ from ours in the Related Work section.
> We also tested SAEs purely as a decomposition tool on extracted activations (as an alternative to SNMF) and report it in Table 2. SNMF performed better, so we use SNMF in the main experiments. In future work, SNMF and SAE could complement each other as alternative post-hoc activation decomposition tools. In our runs, SAE-based decomposition appeared slower to train than SNMF. However, this is only a preliminary observation (not a controlled study).
>
>
>
> We included concern of W2 and Q3 in the introduction.  Thanks for sharing that related work [i]
>
>
>
> **Reference List (copied)**
>
> [a] Label-Free CBM, ICLR 2023
>
> [b] Language-in-a-Bottle, CVPR 2023
>
> [c] DCBM, ICML 2025
>
> [d] CoX-LMM, NeurIPS 2024
>
> [e] CRAFT, NeurIPS 2020
>
> [f] Holistic Concept Extraction, ICML 2022
>
> [g] Sparse Autoencoders Learn Monosemantic Features in Vision-Language Models, arXiv:2504.02821, NeurIPS 2025
>
> [h] Large Multi-modal Models Can Interpret Features in Large Multi-modal Models, arXiv:2411.14982, ICCV 2025
>
> [i] Explaining Similarity in Vision-Language Encoders with Weighted Banzhaf Interactions, arXiv:2508.05430, NeurIPS 2025
>
> [j] Pre-trained Vision-Language Models Learn Discoverable Visual Concepts, arXiv:2404.12652, TMLR 2025

---

> ### Comment · Reviewer_PnqH · 2025-11-24
>
> Thank you for the productive discussion with clarifications, which improved the submission; I increased my rating from 2 to 6.

---

> > ### Author Response · Authors · 2025-11-24
> >
> > Thank you for your thoughtful follow-up and for revising your rating. We’re glad the clarifications addressed your concerns, and we appreciate your constructive feedback throughout.

---

> > > ### Author Response · Authors · 2025-12-01
> > > **Summary comment**
> > >
> > > **We clarified the perceived weaknesses and addressed all questions with additional results in Appendix E.3 and the ablation study in Table 10. As a result, the reviewer was convinced and increased the score from 2 to 6.**

---

### Public Comment · ~Md_Abdul_Kadir1 · 2026-04-13
****Clarification on Review Summary and Score Updates****

## **Clarification on Review Summary**

The paper initially received scores of **2, 6, 6, and 4**. Reviewer **PnqH** raised concerns about **novelty**, **related work comparison**, and **evaluation metrics**.

After the rebuttal, the authors **addressed all concerns**, added **additional results in the appendix**, and **clarified the novelty**. Consequently, **PnqH updated the score from 2 to 6**, as stated in the final comments.

However, due to the **score reset in ICLR 2026**, this update is **not reflected numerically**, and the **meta-review appears to have overlooked it**.

We encourage readers to **review the full discussion and final comments**, not just the meta-review.

---

### Meta-Review · Area_Chair_dv6d · 2026-01-08

**Summary:**

The paper proposes CGDL (Concept-Guided Dictionary Learning), a framework designed to improve the interpretability of Vision-Language Models (VLMs) by extracting sparse and monosemantic visual-text concepts.
The reviewers generally agree that the motivation is convincing. They appreciate the use of contrastive dictionary learning and the two-basis decomposition (concept vs. residual).

However, the reviewers raised several significant concerns that inform a cautious decision:

Novelty and Incremental Contribution: Reviewer PnqH points out that the method is a complex "reusing" of existing components (SAM, CAV, LVLM) and heavily relies on the framework of a specific prior work (CoX-LMM), leading to questions about the independent significance of the contribution.

Methodological Limitations: Reviewer Bj5C notes the reliance on linear decomposition assumptions (which may not capture non-linear reasoning) and the sensitivity of CGDL.

Evaluation Breadth: There is a consensus that the evaluation is somewhat narrow. Specifically, the paper compares primarily against CoX-LMM, lacks a thorough sensitivity analysis of hyperparameters (Reviewer AqDc), and lacks a discussion on computational costs/efficiency (Reviewer wx7H).

Presentation: Some reviewers highlight issues with writing quality.

**Reviewer Concerns:**

Several reviewers' concerns are well-addressed in the rebuttal, such as hyperparameter sensitivity, computational costs/efficiency, and writing quality. Some points are potentially still outstanding: The heavy focus on CoX-LMM as the sole primary baseline (Reviewer PnqH) remains a concern. The authors could design some meaningful baselines for comparison. Reviewer PnqH thinks this method is an over-engineered combination of existing tools (SAM + CAV + LVLM). It is likely difficult for him to change the score. Reviewer Bj5C still has concerns after the rebuttal, e.g., linear decomposition cannot capture abstract or relational concepts in the initial comment.

**Reviewer Scores:**

See Reviewer Concerns

---

> ### Public Comment · ~Md_Abdul_Kadir1 · 2026-04-13
>
> **Due to the score reset in ICLR 2026, this update is not reflected numerically, and the meta-review appears to have overlooked it. We encourage readers to review the full discussion and final comments, not just the meta-review.**

---

### Decision · Program_Chairs · 2026-01-26

Reject